# Experimental evolution for the recovery of growth loss due to genome reduction

**Kenya Hitomi, Yoichiro Ishii, Bei-Wen Ying***

School of Life and Environmental Sciences, University of Tsukuba, Tsukuba, Japan

**Abstract** As the genome encodes the information crucial for cell growth, a sizeable genomic deficiency often causes a significant decrease in growth fitness. Whether and how the decreased growth fitness caused by genome reduction could be compensated by evolution was investigated here. Experimental evolution with an *Escherichia coli* strain carrying a reduced genome was conducted in multiple lineages for approximately 1000 generations. The growth rate, which largely declined due to genome reduction, was considerably recovered, associated with the improved carrying capacity. Genome mutations accumulated during evolution were significantly varied across the evolutionary lineages and were randomly localized on the reduced genome. Transcriptome reorganization showed a common evolutionary direction and conserved the chromosomal periodicity, regardless of highly diversified gene categories, regulons, and pathways enriched in the differentially expressed genes. Genome mutations and transcriptome reorganization caused by evolution, which were found to be dissimilar to those caused by genome reduction, must have followed divergent mechanisms in individual evolutionary lineages. Gene network reconstruction successfully identified three gene modules functionally differentiated, which were responsible for the evolutionary changes of the reduced genome in growth fitness, genome mutation, and gene expression, respectively. The diversity in evolutionary approaches improved the growth fitness associated with the homeostatic transcriptome architecture as if the evolutionary compensation for genome reduction was like all roads leading to Rome.

**\*For correspondence:**
ying.beiwen.gf@u.tsukuba.ac.jp

**Competing interest:** The authors declare that no competing interests exist.

## eLife assessment

This is an **important** study of the recovery of genome-reduced bacterial cells in laboratory evolution experiments to understand how they regain their fitness. Through the analysis of gene expression and a series of tests, the authors present **convincing** evidence indicating distinct molecular changes in the evolved bacterial strains, although the precise mechanisms remain uncharacterized. These findings imply that diverse mechanisms are employed to offset the effects of a reduced genome, offering intriguing insights into genome evolution.

## Introduction

The genome encodes the information for cell growth, and its size is likely the evolutionary consequence (*Koonin, 2009*; *Lynch, 2006*; *Lynch and Conery, 2003*). To determine the essential genomic sequence of modern cells, removing redundant DNA sequences from the wild-type genome in bacteria, so-called genome reduction, has been challenged to a large extent (*Kotaka et al., 2023*; *Kato and Hashimoto, 2007*; *Pósfai et al., 2006*; *Hashimoto et al., 2005*). These efforts have been made to discover the minimal genetic requirement for a free-living organism growing under the defined conditions (*Aida and Ying, 2023*; *Breuer et al., 2019*; *Hutchison et al., 2016*). It resulted in the finding of the coordination of genome with cell growth, that is, genome reduction significantly decreased the growth rate of *Escherichia coli* cells independent of culture media or growth forms (*Hitomi et al., 2022*; *Xue et al., 2021*; *Kurokawa et al., 2016*). Slow growth or fitness decline was

commonly observed in the genetically reduced (*Kurokawa et al., 2016*; *Karcagi et al., 2016*) and chemically synthesized genomes (*Hutchison et al., 2016*; *Gibson et al., 2010*).

The growth decrease could be recovered by experimental evolution. Growth fitness generally represents the adaptiveness of the living organism to the defined environment (*Orr, 2009*). Although the reduced genomes somehow showed differentiated evolvability compared to the wild-type genomes (*Nishimura et al., 2017*; *Csörgo et al., 2012*; *Umenhoffer et al., 2010*), their evolutionary adaptation to the environmental changes has been successfully achieved (*Kurokawa et al., 2022*; *Choe et al., 2019*; *Suzuki et al., 2014*). Experimental evolution under the defined culture condition successfully increased the decreased growth rates of the reduced genomes (*Nishimura et al., 2017*; *Kurokawa et al., 2022*; *Choe et al., 2019*) and fastened the slow-growing synthetic genome (*Moger-Reischer et al., 2023*). The evolutionary rescue of the growth rate must be associated with the changes in genomic sequence and gene expression benefited for the growth fitness, as what happened in nature adaptive evolution (*Ishikawa et al., 2017*; *Rozen et al., 2002*; *Lynch, 2010*). Our previous study showed that the decreased growth rate caused by the absence of a sizeable genomic sequence could be complemented by introducing the mutations elsewhere in the genome (*Kurokawa et al., 2022*; *Choe et al., 2019*). Additionally, the changes in growth rate caused by either genome reduction or experimental evolution were dependent on the genome size but not the specific gene function (*Kurokawa et al., 2022*).

A genome-wide understanding of the evolutionary compensated fitness increase of the reduced genome is required. The experimental evolution compensated for the genome reduction-mediated growth changes was considered stringently related to transcriptome reorganization. Previous studies have identified conserved features in transcriptome reorganization, despite significant disruption to gene expression patterns resulting from either genome reduction or experimental evolution (*Matsui et al., 2023*; *Nagai et al., 2020*; *Ying and Yama, 2018*). The findings indicated that experimental evolution might reinstate growth rates that have been disrupted by genome reduction to maintain homeostasis in growing cells. Although the reduced growth rate caused by genome reduction could be recovered by experimental evolution, it remains unclear whether such an evolutionary improvement in growth fitness was a general feature of the reduced genome and how the genome-wide changes occurred to match the growth fitness increase. In the present study, we performed the experimental evolution with a reduced genome in multiple lineages and analyzed the evolutionary changes of the genome and transcriptome.

## Results
### Fitness recovery of the reduced genome by experimental evolution
Experimental evolution of the reduced genome was conducted to regain the growth fitness, which was decreased due to genome reduction. The *E. coli* strain carrying a reduced genome, derived from the wild-type genome W3110, showed a significant decline in its growth rate in the minimal medium compared to the wild-type strain (*Kurokawa et al., 2016*). To improve the genome reduction-mediated decreased growth rate, the serial transfer of the genome-reduced strain was performed with multiple dilution rates to keep the bacterial growth within the exponential phase (*Figure 1—figure supplement 1*), as described (*Nishimura et al., 2017*; *Kurokawa et al., 2022*). Nine evolutionary lineages were conducted independently (*Figure 1—figure supplement 2*, *Supplementary file 1*). A gradual increase in growth rate was observed along with the generation passage, which was combined with a rapid increase in the early evolutionary phase and a slow increase in the later phase (*Figure 1A*). As the growth increases were calculated according to the initial and final records, the exponential growth rates of the ancestor and evolved populations were obtained according to the growth curves for a precise evaluation of the evolutionary changes in growth. The results demonstrated that most evolved populations (Evos) showed improved growth rates, in which eight out of nine Evos were highly significant (*Figure 1B*, upper panel). However, the magnitudes of growth improvement were considerably varied, and the evolutionary dynamics of the nine lineages were somehow divergent (*Figure 1—figure supplement 2*). It indicated the diversity in the evolutionary approaches for improved fitness. Eight of nine Evos achieved faster growth than the genome-reduced ancestor (Anc), whereas all Evos decreased in their saturated population densities (*Figure 1B*, bottom panel). In comparison to the wild-type strain carrying the full-length genome (WT), the primarily decreased growth rate caused by

genome reduction was significantly improved by experimental evolution (*Figure 1C*, upper panel), associated with the considerable reduction in saturated density (*Figure 1C*, bottom panel). It demonstrated that the experimental evolution could compensate for the genome reduction with a trade-off in population size (carrying capacity), consistent with the previous findings (*Kurokawa et al., 2022*; *Choe et al., 2019*).

Intriguingly, a positive correlation was observed between the growth fitness and the carrying capacity of the Evos (*Figure 1D*). It was consistent with the positive correlations between the colony growth rate and the colony size of a genome-reduced strain (*Hitomi et al., 2022*) and between the growth rates and the saturated population size of an assortment of genome-reduced strains (*Kurokawa et al., 2016*). Nevertheless, the negative correlation between growth rate and carrying capacity, known as the *r/K* selection (*Engen and Saether, 2017*; *Luckinbill, 1978*), was often observed as the trade-off relationship between *r* and *K* in the evolution and ecology studies (*Marshall et al., 2023*; *Molenaar et al., 2009*; *Wortel et al., 2018*). As the *r/K* trade-off was proposed to balance the cellular metabolism that resulted from the cost of enzymes involved (*Wortel et al., 2018*), the deleted genes might play a role in maintaining the metabolism balance for the *r/K* correlation. On the other hand, the experimental evolution (i.e., serial transfer) was strictly performed within the exponential growth phase; thus, the evolutionary selection was supposed to be driven by the growth rate without selective pressure to maintain the carrying capacity. The declined carrying capacity might have been its neutral 'drift' but not a trade-off to the growth rate. Independent and parallel experimental evolution of the reduced genomes selecting either *r* or *K* is required to clarify the actual mechanisms.

## Significant variation and random localization of genome mutations

Genome resequencing (*Supplementary file 2*) identified a total of 65 mutations fixed in the nine Evos (*Table 1*). The number of mutations largely varied among the nine Evos, from 2 to 13, and no

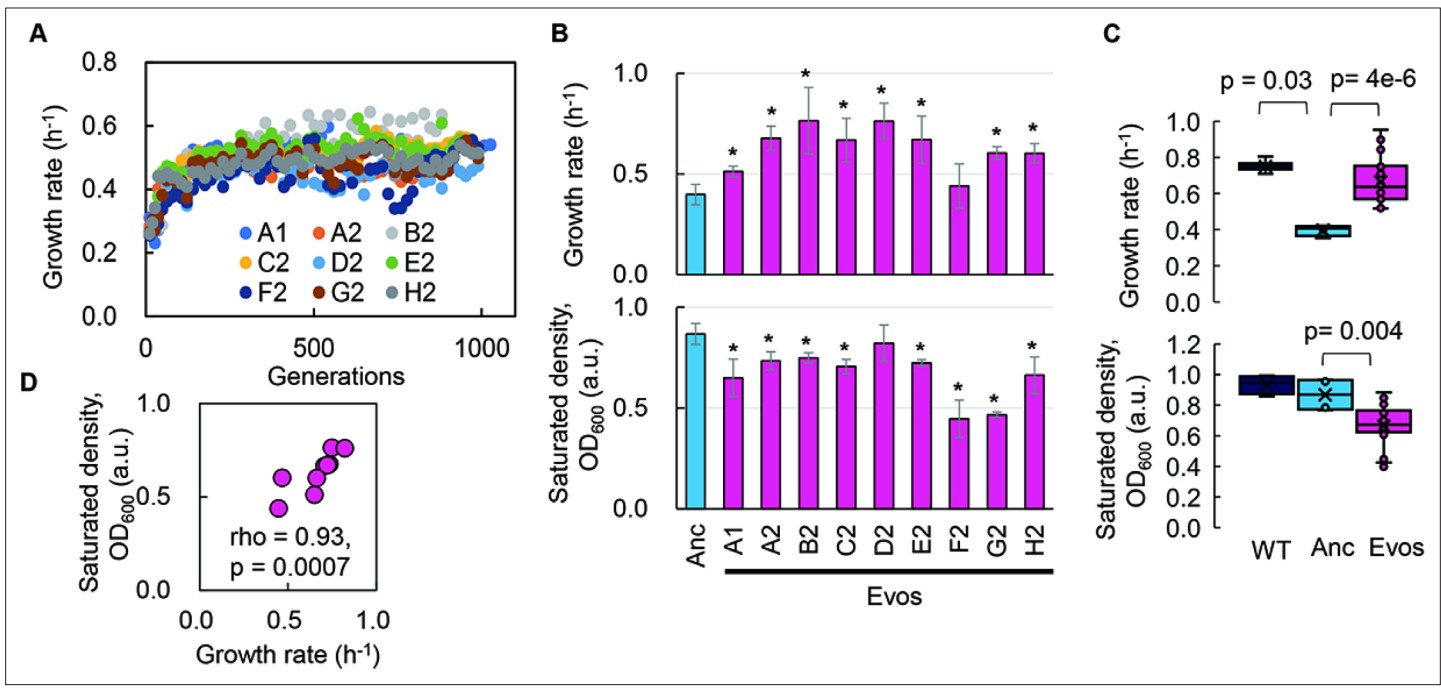

**Figure 1.** Fitness increase of the reduced genome mediated by experimental evolution. (**A**) Temporal changes in growth rate. Color variation indicates the nine evolutionary lineages. (**B**) Growth rate and maximal population size of the reduced genome. Blue and pink indicate the common ancestor and the nine evolved populations, respectively. Standard errors are shown according to the biological replications (N = 4–6). (**C**) Boxplots of growth rate and maximum. Cross and open circles indicate the mean and individual values, respectively. Statistical significance evaluated by Mann–Whitney *U* tests is indicated. (**D**) Correlation between growth rate and maximum. Spearman's rank correlation coefficient and p-value are indicated.

The online version of this article includes the following figure supplement(s) for figure 1:

**Figure supplement 1.** Temporal OD$_{600}$ records of the overnight cultures.

**Figure supplement 2.** Temporal changes in growth rate during experimental evolution.

common mutation was detected in all nine Evos (*Supplementary file 3*). A 1199 bp deletion of *insH* was frequently found in the Evos (*Supplementary file 3*, highlighted), which agreed with its function as a transposable sequence. Also, 51 out of 65 mutations occurred in the genes, and 45 out of 51 were SNPs. As 36 out of 45 SNPs were nonsynonymous, which was highly significant compared to random mutations ($p<0.01$), the mutated genes might benefit fitness increase. In addition, the abundance of mutations was unlikely to be related to the magnitude of fitness increase. There was no significant correlation between the number of mutations and the growth rate in a statistical view ($p>0.1$). Even from an individual close-up viewpoint, the abundance of mutations poorly explained the fitness increase. For instance, A2 accumulating only two mutations presented a highly increased growth rate compared to F2 of eight mutations (*Table 1*), which poorly improved the growth (*Figure 1B*). B2, D2, and E2 all succeeded in increasing fitness to an equivalent degree (*Figure 1B*), whereas they fixed 13, 7, and 3 mutations, respectively (*Table 1*). The mutated genes were varied in 14 gene categories, somehow more frequently in the gene categories of Transporter, Enzyme, and Unknown function (*Supplementary file 4*). They seemed highly related to essentiality (*Hashimoto et al., 2005*; https://shigen.nig.ac.jp/ecoli/pec/genes.jsp) as 11 out of 49 mutated genes were essential (*Supplementary file 3*). Although the essentiality of genes might differ between the wild-type and reduced genomes, the experimentally determined 302 essential genes in the wild-type *E. coli* strain were used for the analysis, of which 286 were annotated in the reduced genome. The ratio of essential genes in the mutated genes was significantly higher than that in the total genes (286 out of 3290 genes, chi-square test $p=0.008$). As the essential genes were determined according to the growth (*Baba et al., 2006*) and were known to be more conserved than nonessential ones (*Jordan et al., 2002*; *Zhang, 2022*), the high frequency of the mutations fixed in the essential genes was highly intriguing and reasonable. The large variety of genome mutations fixed in the independent lineages might result from a highly rugged fitness landscape (*Van Cleve and Weissman, 2015*). Nevertheless, it was unclear whether and how these mutations were explicitly responsible for recovering the growth rate of the reduced genome.

Additionally, there were no overlapping genomic positions for the 65 mutations in the nine Evos (*Figure 2A*). Random simulation was performed to verify whether there was any bias or hotspot in the genomic location for mutation accumulation due to the genome reduction. A total of 65 mutations were randomly generated on the reduced genome (*Figure 2B*), and the genomic distances from the mutations to the nearest genome reduction-mediated scars were calculated. Welch's *t*-test was performed to evaluate whether the genomic distances calculated from random mutations significantly differed from those from the mutations accumulated in the Evos. As the mean of p-values (1000 times of random simulations) was insignificant (*Figure 2C*, $\mu_p > 0.05$), the mutations fixed on the reduced

**Table 1.** Overview of fixed genome mutations.
The number of mutations in the nine Evos is shown separately and summed. SNP, N, and S indicate single nucleotide substitution, nonsynonymous, and synonymous SNP, respectively.

| Evos | All | Intergenic | Genic | | Genic SNP | |
| | | | Indel | SNP | N | S |
|------|-----|-----------|-------|-----|---|---|
| A1 | 3 | 0 | 0 | 2 | 1 | 1 |
| A2 | 2 | 1 | 0 | 1 | 1 | 0 |
| B2 | 13 | 0 | 1 | 11 | 9 | 2 |
| C2 | 11 | 1 | 0 | 9 | 8 | 1 |
| D2 | 7 | 0 | 1 | 5 | 4 | 1 |
| E2 | 3 | 1 | 0 | 1 | 1 | 0 |
| F2 | 8 | 1 | 0 | 6 | 5 | 1 |
| G2 | 12 | 2 | 2 | 6 | 4 | 2 |
| H2 | 6 | 0 | 2 | 4 | 3 | 1 |
| Sum | 65 | 6 | 6 | 45 | 36 | 9 |

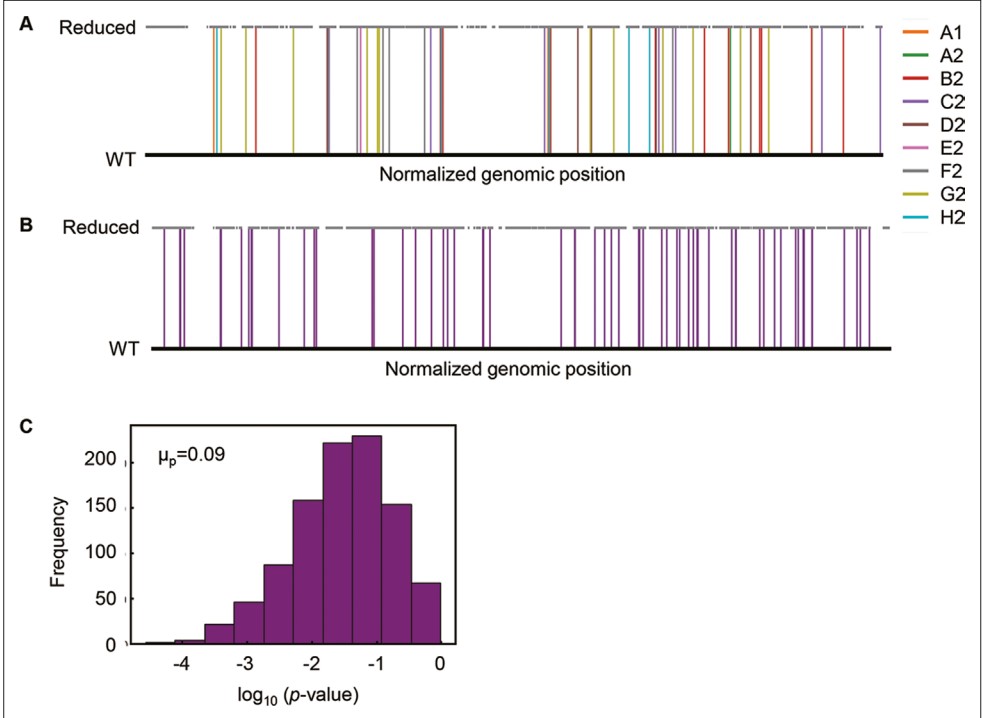

**Figure 2.** Genomic localization of mutations. (**A**) Normalized genomic positions of all mutations. The vertical lines highlight the total 65 mutations fixed in the nine Evos. Color variation indicates the nine Evos. WT and reduced represent the wild-type and reduced genomes used in the present study. (**B**) Normalized genomic positions of random mutations. The simulation of 65 mutations randomly fixed in the reduced genome was performed 1000 times. As an example of the simulation, the genomic positions of 65 random mutations are shown. The vertical lines in purple indicate the mutations. (**C**) Statistical significance of the genome locational bias of mutations. The distance from the mutated location to the nearest genomic scar caused by genome reduction was calculated. The mutations accumulated in the nine Evos and the 1000-time random simulation were all subjected to the calculation. The significance of genome locational bias of the mutations in Evos was evaluated by Welch's *t*-test. The histogram of 1000 tests for 1000 simulated results is shown. The mean of p-values ($\mu_p$) is indicated, which is within the 95% confidence interval ($0.07 < \mu_p < 0.09$).

genome were either closer or farther to the genomic scars, indicating that there was no locational bias for mutation accumulation caused by genome reduction.

## Common evolutionary direction and homeostasis in transcriptome reorganization

Since no specificity was detected in the genome mutations, whether these mutations disturbed the genome-wide gene expression pattern was investigated. Hierarchical clustering and principal component analysis (PCA) showed that the evolved transcriptomes were directed to similar patterns but divergent to the WT transcriptome (*Figure 4—figure supplement 1*). The evolutionary direction of the transcriptomes of the reduced genome was not approaching the wild-type transcriptome. As the *E. coli* chromosome was structured, whether the genome reduction caused the changes in its architecture, which led to the differentiated transcriptome reorganization in the Evos, was investigated. The chromosomal periodicity of gene expression was analyzed to determine the structural feature of genome-wide pattern, as previously described (*Nagai et al., 2020*; *Ying et al., 2013*). The analytical results showed that the transcriptomes of all Evos presented a common six-period with statistical significance, equivalent to those of the wild-type and ancestral reduced genomes (*Figure 3A*, *Supplementary file 5*). It demonstrated that the chromosomal architecture of gene expression patterns remained highly conserved, regardless of the considerably varied mutations. The homeostatic periodicity was consistent with our previous findings that the chromosomal periodicity of the transcriptome was independent of genomic or environmental variation (*Matsui et al., 2023*; *Nagai et al., 2020*).

In addition, the genomic locations of the mutations seemed irrelevant to chromosomal periodicity (*Figure 3A*, red lines). No mutagenesis hotspot was observed even if these mutations were accumulated on a single genome (*Figure 2A*). Alternatively, the expression levels of the mutated genes were somehow higher than those of the remaining genes (*Figure 3B*). As the regulatory mechanisms or the gene functions were supposed to be disturbed by the mutations, the expression levels of individual genes might have been either up- or downregulated. Nevertheless, the overall expression levels of all mutated genes tended to be increased. One of the reasons was assumed to be the mutation essentiality, which remained to be experimentally verified. The ratio of the essential genes in the mutated genes was ~22% (*Supplementary file 3*), much higher than the ratio (~9%) of essential to all genes (286 out of 3290) in the reduced genome. As the essential genes showed higher expression levels than the nonessential ones (*Ying et al., 2017*), the high essentiality of the mutated genes might result in a higher mean expression level. On the other hand, the high frequency of the mutations fixed in the essential genes was unexpected because the essential genes were generally more conserved than nonessential ones (*Jordan et al., 2002*). The essentiality of the mutations might have participated in maintaining the homeostatic transcriptome architecture of the reduced genome.

## Diversified functions and pathways of the differentially expressed genes

As the evolved transcriptomes were differentiated from those of the WT and reduced genomes (*Figure 4—figure supplement 1*), the differentially expressed genes (DEGs) were further identified to discover the gene functions or biological processes contributing to the fitness changes. Instead of an in-depth survey on the directional changes of the DEGs, the abundance and functional enrichment of DEGs were investigated to achieve an overview of how significant the genome-wide fluctuation in gene expression, which ignored the details of individual genes. The abundance of DEGs among the

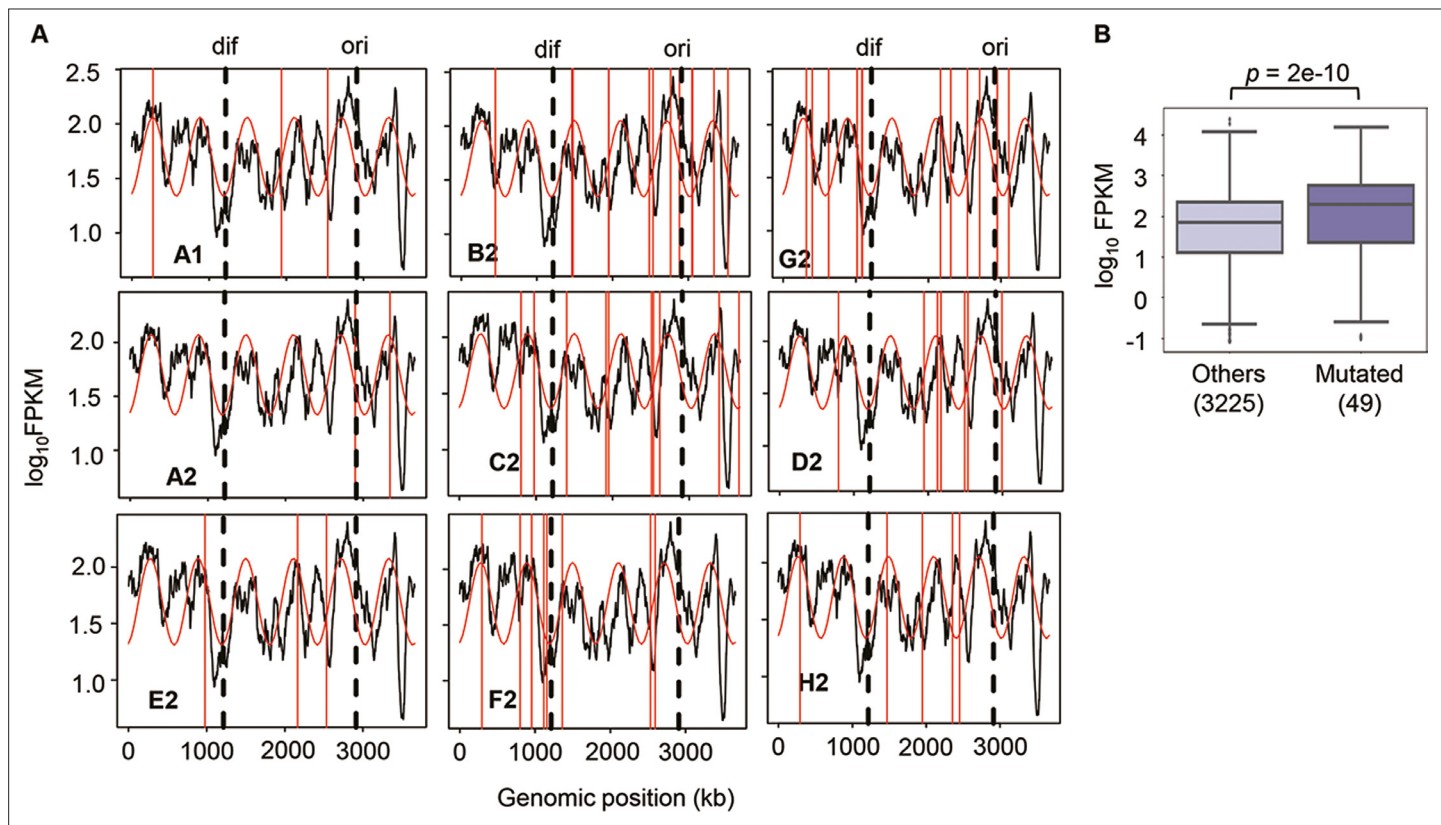

**Figure 3.** Chromosomal periodicity of transcriptome and mutated gene expression. (**A**) Chromosomal periodicity of transcriptomes. The transcriptomes of the nine Evos are shown. Black lines, red curves, and red vertical lines indicate the gene expression levels, fitted periods, and locations of mutations, respectively. *Ori* and *dif* are indicated with the vertical broken lines. (**B**) Boxplot of gene expression levels. Gene expression levels of the 49 mutated genes in the nine Evos and the remaining 3225 genes are shown. Statistical significance evaluated by Welch's *t*-test is indicated.

Evos varied from 333 to 1130 genes and of few overlaps (*Figure 4A*). Most DEGs were unique to each evolutionary lineage, and the common DEGs across all Evos were only 108 genes (*Supplementary file 6*). The number of DEGs partially overlapped among the Evos declined along with the increased lineages of Evos (*Figure 4B*). Enrichment analysis showed that only the histidine-related pathways were significantly enriched in the common DEGs (*Figure 4C*). Functional enrichment of the DEGs in individual Evos showed that the amino acid metabolism considerably participated and that the enriched pathways were poorly overlapped (*Figure 4—figure supplement 2*). These findings strongly suggested no universal rule for evolutionary changes of the reduced genome.

In comparison, 1226 DEGs were induced by genome reduction. The common DEGs of genome reduction and evolution varied from 168 to 540, fewer than half of the DEGs responsible for genome reduction in all Evos (*Figure 5A*). The conclusion remained consistent even if the DEGs were determined with an alternative method, RankProd (*Figure 5—figure supplement 1*). Functional enrichment of the DEGs observed the specific transcriptional regulation and metabolic pathways participating in the transcriptome reorganization in response to genome reduction and evolution. Only σ38 was enriched in the genome reduction-mediated DEGs, the only regulon partially overlapping with the Evos (*Figure 5B*). No regulon in common was enriched, besides a few partially overlapped regulons, that is, GadW, GadX, and RcsB (*Figure 5B*). In addition, both the number of enriched pathways and their overlaps were significantly differentiated among the nine Evos or between Evos and reduced genomes (*Figure 5C*). No common pathways were commonly enriched between the genome reduction-mediated DEGs and Evos; no matter whether annotated the metabolic pathways with Gene Ontology (GO) or Kyoto Encyclopedia of Genes and Genomes (KEGG) (*Figure 5—figure supplement 2*). The flagellar assembly, the only enriched pathway in genome reduction-mediated DEGs, was absent in all Evos (*Figure 5D*). Alternatively, the amino acids-related metabolisms were frequently detected in the Evos, for example, histidine metabolism, biosynthesis of amino acids, etc. (*Figure 5D*, *Figure 5—figure supplement 2*). The variable pathways in the Evos indicated that evolution compensated for the genome reduction in various ways, which differed from how the genome reduction was caused.

## Gene modules responsible for the evolutionary changes of the reduced genome

Genome mutation analysis and transcriptome analysis failed to identify the common gene categories or pathways that correlated to the evolution of the reduced genome; thus, the gene modules correlated to evolution were newly evaluated. The weighted gene co-expression network analysis (WGCNA) (*Langfelder and Horvath, 2008*) was performed toward the evolved transcriptomes as tested previously (*Matsui et al., 2023*). Reconstruction of the 3290 genes in the reduced genome led to 21 gene modules, comprising 8–320 genes per module (*Figure 6—figure supplement 1*). Hierarchical clustering of these modules showed that roughly three major classes could be primarily divided (*Figure 6A*). Functional correlation analysis showed that three of 21 modules (M2, M10, and M16) were highly significantly correlated to the growth fitness, the number of DEGs, and mutation frequency, respectively (*Figure 6A*). These modules M2, M10, and M16 might be considered the hotspots for the genes responsible for growth fitness, transcriptional reorganization, and mutation accumulation of the reduced genome in evolution, respectively. It indicated that the three modules were highly essential and functionally differentiated for growth control, transcriptional change, and mutagenesis.

Enrichment analyses further identified the gene categories, regulons, and metabolisms that significantly appeared in the three modules (*Figure 6B*). Two gene categories of nonessential function were enriched in M10, and the module correlated to the number of DEGs; in contrast, no gene category was enriched in M2 and M16 (*Figure 6B*, left). All three modules successfully enriched the regulons without overlaps (*Figure 6B*, middle). It indicates that the main regulatory mechanisms participating in the three modules for growth control, transcriptional change, and mutagenesis were divergent. GO enrichment resulted in various biological processes in the three modules, roughly related to transport, transposition, and translation in M2, M10, and M16, respectively (*Figure 6B*, right). Compared to the enriched functions of the genes that disappeared due to genome reduction (*Figure 6B*, bottom), the gene categories of phage and unknown function and the biological processes related to DNA transposition and integration were commonly identified in M10. It strongly indicated that M10 was

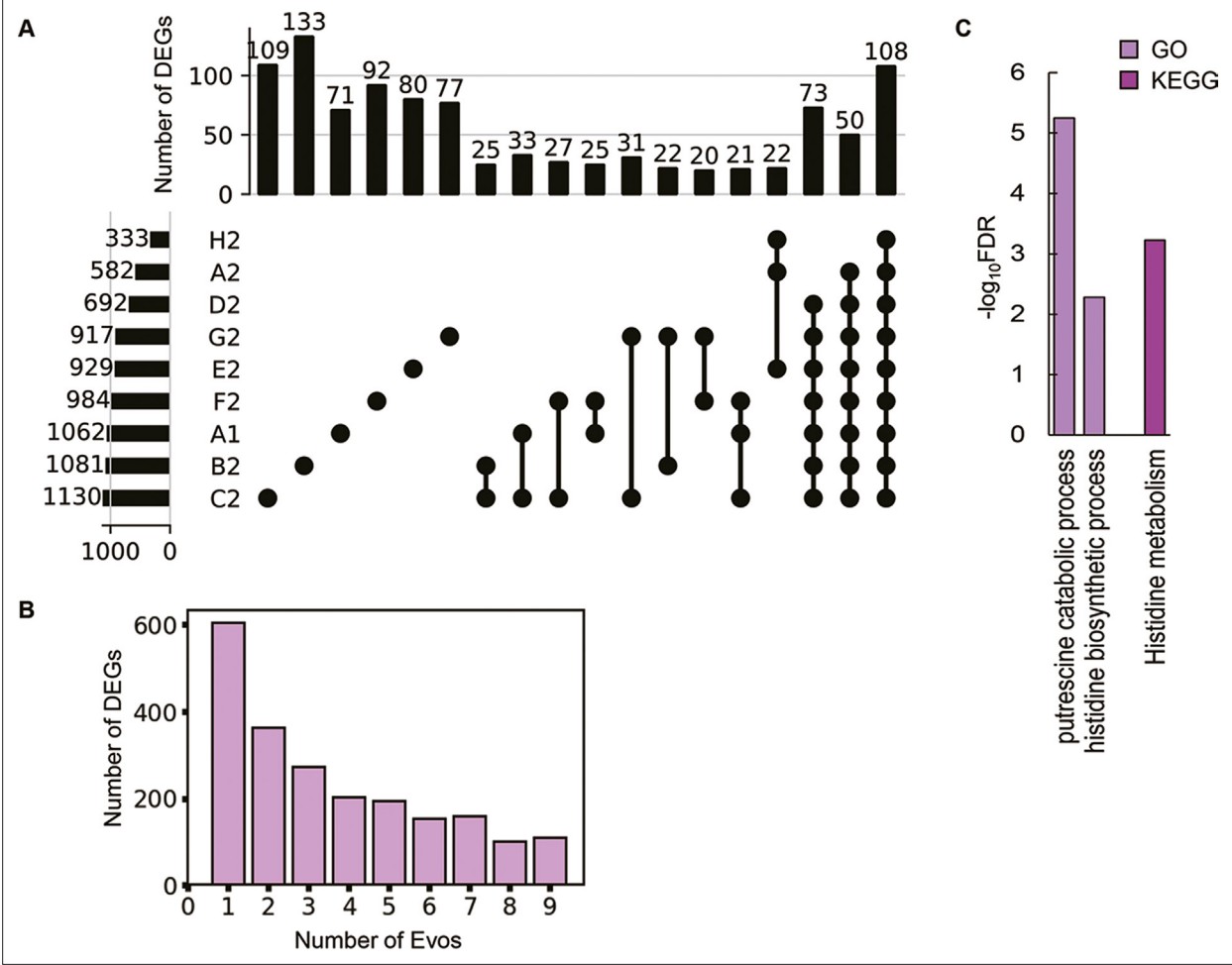

**Figure 4.** Differentially expressed genes (DEGs) and their enriched functions. (**A**) Commonality of DEGs in the nine Evos. Closed circles represent the combinations of the Evos. Vertical and horizontal bars indicate the number of the overlapped DEGs in the combinations and the number of all DEGs in each Evo, respectively. The combinations with more than 20 DEGs in common are shown. (**B**) The number of DEGs overlapped among the Evos. The numbers of DEGs overlapped across 2–9 Evos are shown. The number of Evos detected in the single Evo is indicated as 1. (**C**) Enriched function in common. The Kyoto Encyclopedia of Genes and Genomes (KEGG) and Gene Ontology (GO) terms enriched in the common DEGs across the nine Evos are shown. The statistical significance (false discovery rate [FDR]) of the enriched pathways and biological processes is shown on a logarithmic scale represented by color gradation.

The online version of this article includes the following figure supplement(s) for figure 4:

**Figure supplement 1.** Clustering and principal component analysis of transcriptomes.

**Figure supplement 2.** Enriched functions of differentially expressed genes (DEGs) in Evos.

responsible for genome reduction. The newly constructed gene networks successfully identified three modules correlated to mutation, DEGs, and growth, revealing the functional differentiation responsible for evolution to maintain the homeostatic transcriptome architecture for a growing cell.

## Discussion

The evolutionary compensation for genome reduction was directed toward an identical goal of increased fitness but differentiated in the manner of genomic changes. Firstly, various genetic functions seemed to trigger the increased growth rates of the Evos. A few mutations could compensate for the sizeable genomic deficiency. In particular, mutations in the essential genes, such as RNA polymerases (*rpoA*, *rpoB*, *rpoD*) identified in three Evos (*Supplementary file 3*), were supposed to participate in the global regulation for improved growth. Nevertheless, the considerable variation in the fixed mutations without overlaps among the nine Evos (*Table 1*) implied no common mutagenetic strategy

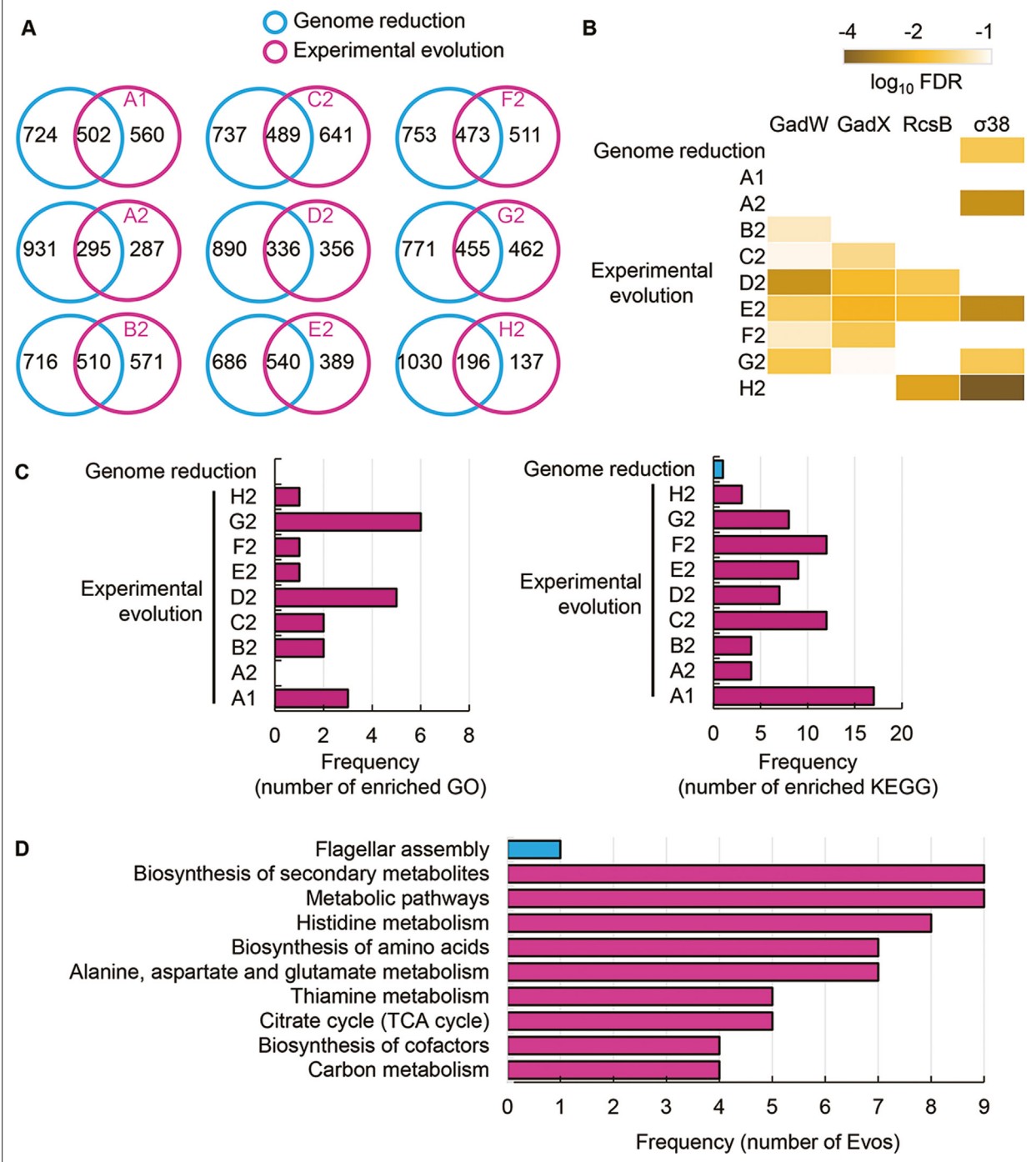

**Figure 5.** Transcriptome comparison between genome reduction and evolution. (**A**) Venn diagrams of differentially expressed genes (DEGs) induced by genome reduction and evolution. The number of individual and overlapped DEGs is indicated. (**B**) Heatmap of enriched regulons. Statistically significant regulons are shown with the false discovery rate (FDR) values on a logarithmic scale. (**C**) Number of enriched functions in common. Left and right panels indicate the numbers of enriched Gene Ontology (GO) terms and Kyoto Encyclopedia of Genes and Genomes pathways caused by genome reduction and evolution, respectively. (**D**) Enriched functions in common. The overlapped GO terms enriched in the nine Evos and genome reduction are shown. Blue and pink represent genome reduction and evolution, respectively.

The online version of this article includes the following figure supplement(s) for figure 5:

**Figure supplement 1.** Venn diagrams of the differentially expressed genes (DEGs) determined by RankProd.

**Figure supplement 2.** Heatmap of the significance of enriched functions.

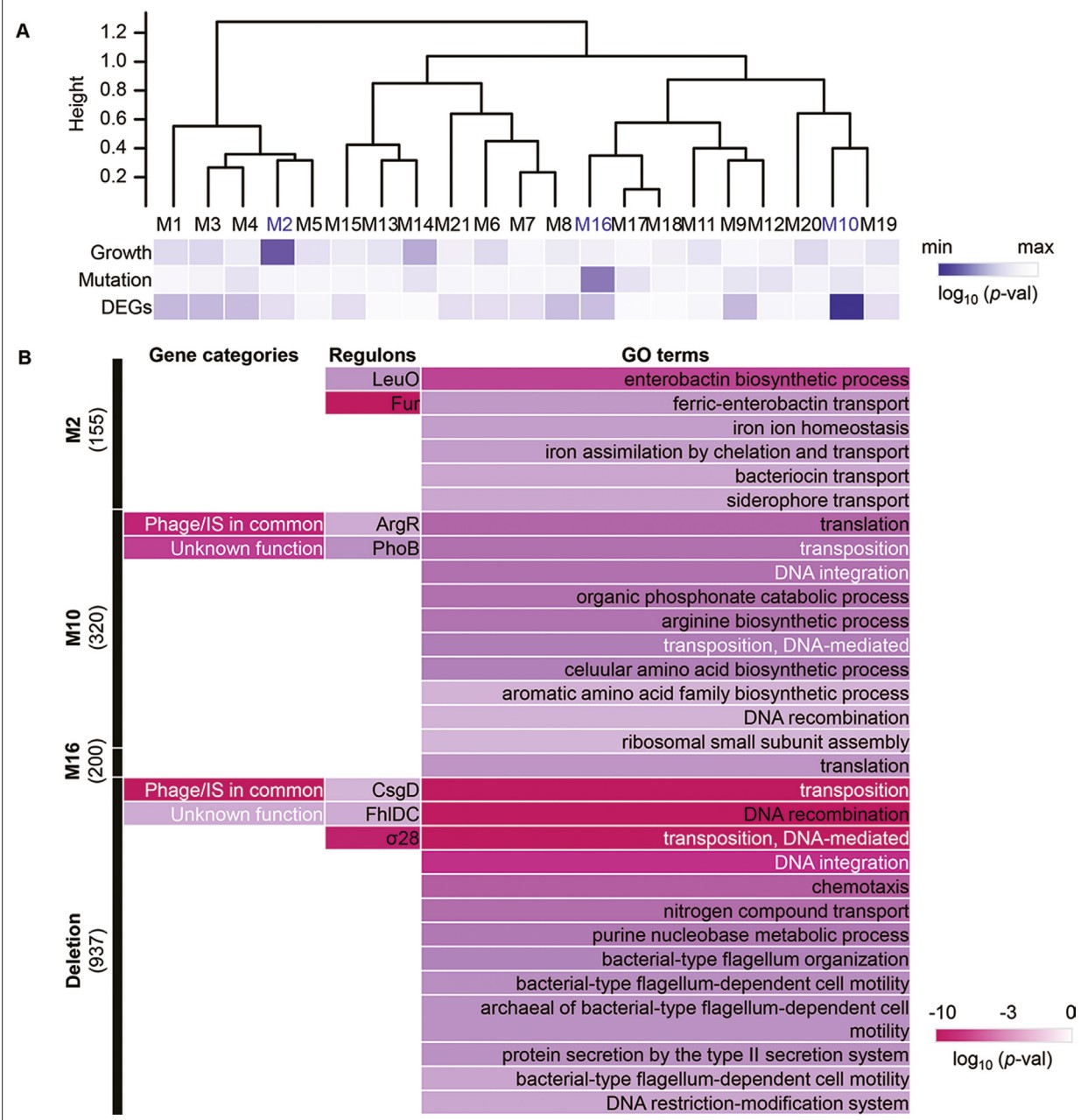

**Figure 6.** Reconstructed gene modules. (**A**) Cluster dendrogram of the gene modules reconstructed by weighted gene co-expression network analysis (WGCNA). A total of 21 gene modules (M1–M21) were reconstructed. The significance of the correlation coefficients of the gene modules to growth, mutation, and expression is represented in purple gradation. From light to dark indicates the logarithmic p-values from high to low. (**B**) Enriched functions of gene modules and deletion. Enriched gene categories, regulons, and GO terms are shown from left to right. The number of the genes assigned in the three gene modules and the genomic deletion for genome reduction is indicated in the brackets. Color gradation indicates the normalized p-values on a logarithmic scale.

The online version of this article includes the following figure supplement(s) for figure 6:

**Figure supplement 1.** Reconstruction of gene modules according to transcriptomes.

for the evolutionary improvement of growth fitness. It was supported by the fact that no genomic locational bias for mutations fixed in the evolution (*Figure 2*). Secondly, the transcriptomes presented conserved chromosomal architectures (*Figure 3*) and universal directional changes (*Figure 4—figure supplement 1*), regardless of the significant and differentiated changes in gene expression in response to evolution (*Figure 4*, *Figure 4—figure supplement 2*). Although the periodicity of chromosomal

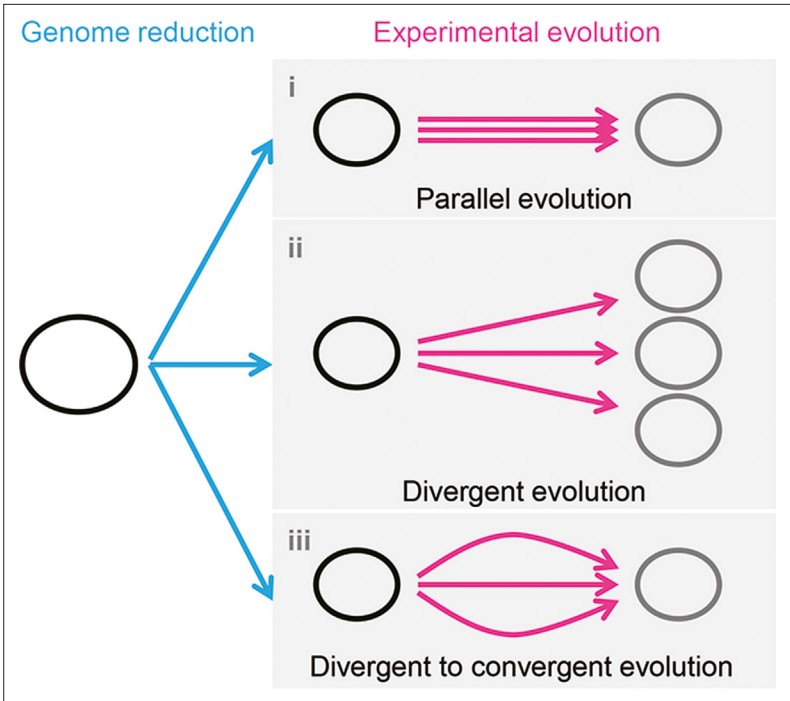

**Figure 7.** Schematic drawing of evolutionary approaches for the reduced genome. Three evolutionary strategies are proposed. Pink and blue arrowed lines indicate experimental evolution and genome reduction, respectively. The size of the open cycles represents the genome size. Black and gray indicate the ancestor and evolved genomes, respectively.

architecture was well known (***Mathelier and Carbone, 2010***; ***Krogh et al., 2019***) and coordinated with the bacterial growth rate (***Nagai et al., 2020***; ***Liu et al., 2020***), employing its homeostasis as the evolutionary consequence provided a conceptual and unique understanding of growing cells.

It is unclear whether the differentiation in evolutionary paths was particularly significant for the reduced genome used in the present study. Evolution studies often focus on finding the common mutations accumulated in multiple evolutionary lineages to obtain the reasonable mechanism responsible for the adaptation to the defined condition. Common mutations (***Suzuki et al., 2014***) or identical genetic functions (***Lu et al., 2022***) were reported in the experimental evolution with different reduced genomes, commonly known as parallel evolution (***Figure 7***, i). In addition, as not all mutations contribute to the evolved fitness (***Suzuki et al., 2014***; ***Lu et al., 2022***), another strategy for varied phenotypes was known as divergent evolution (***Figure 7***, ii). The present study accentuated the variety of mutations fixed during evolution. Considering the high essentiality of the mutated genes (***Supplementary file 3***), most or all mutations were assumed to benefit the fitness increase, partially demonstrated previously (***Kurokawa et al., 2022***). Nevertheless, the evolved transcriptomes presented a homeostatic architecture, revealing the divergent to convergent evolutionary strategy (***Figure 7***, iii). Multiple evolutionary paths for the reduced genome to improve growth fitness were likely all roads leading to Rome.

In addition, the transcriptome reorganization for fitness increase triggered by evolution differed from that for fitness decrease caused by genome reduction. General analyses failed to detect the regulatory network or genetic function mediated by genome reduction and evolution in common. Instead, the newly constructed gene modules successfully enriched the gene categories of mobile elements and unknown functions (***Figure 6B***, left) as the evolutionary compensation for genome reduction. The represented mobile elements, flagella, were known to be responsive to environmental stresses such as hypoosmotic pressure or pH (***Ikeda et al., 2020***; ***Maurer et al., 2005***). Genome reduction and evolution seemed equivalent to the stress response in *E. coli*. These findings were reasonable as enterobactin protected *E. coli* from oxidative stress (***Adler et al., 2014***; ***Peralta et al., 2016***), and enterobactin biosynthesis was upregulated for biofilm formation in genome-reduced *E. coli* (***May and Okabe, 2011***). The compensation of evolution to genome reduction not only verified

the known function and mechanism from a global regulatory viewpoint but also revealed a novel understanding of the molecular mechanisms and gene functions.

The discriminated functions of gene modules might play a crucial role in response to genomic and evolutionary changes. WGCNA was conducted to discover the potential correlation of gene expression to growth fitness. It succeeded in finding the genes participating in the evolutionary changes of the reduced genome to regain growth fitness. Three enriched gene modules were assumed separately responsible for replication, transcription, and population dynamics (*Figure 6B*). The growth-correlated gene module significantly enriched the iron-related biological functions (M2). Although the translation was commonly reported to be correlated to the growth rate (*Scott and Hwa, 2023*; *Dai et al., 2016*), it was enriched in the gene module coordinated to transcriptional changes (M10). Such functional differentiation of the gene modules might connect with the differentiated medium components responsible for varied bacterial growth phases, which was observed using the high-throughput growth assay in hundreds of medium combinations combined with machine learning (*Aida et al., 2022*; *Ashino et al., 2019*). We assumed that the various chemicals disturbed different metabolic fluxes in which different gene modules might have participated. The biological meaningfulness of the gene modules suggested an alternative genetic classification besides the commonly used clustering criteria, such as Gene Orthology (*Ashburner et al., 2000*) and Regulon (*Salgado et al., 2013*). In summary, the present study provided a representative example showing multiple evolutionary paths (i.e., gene mutation and expression) directed the reduced genome to the improved fitness with the homeostatic transcriptome.

## Materials and methods

### *E. coli* strains and culture medium

The *E. coli* K-12 W3110 wild-type and its genome-reduced strains were initially distributed by the National BioResource Project of the National Institute of Genetics. The reduced genome has been constructed by multiple deletions of large genomic fragments (*Mizoguchi et al., 2008*), which led to an approximately 21% smaller size than its parent wild-type genome W3110. The minimal medium M63 was used as described in detail elsewhere (*Kurokawa et al., 2016*; *Kurokawa and Ying, 2017*). In brief, the medium contains 62 mM dipotassium hydrogen phosphate, 39 mM potassium dihydrogen phosphate, 15 mM ammonium sulfate, 15 µM thiamine hydrochloride, 1.8 µM iron (II) sulfate, 0.2 mM magnesium sulfate, and 22 mM glucose.

### Experimental evolution

The genome-reduced *E. coli* strain was evolved in 2 ml of the M63 medium by serial transfer, as previously described (*Nishimura et al., 2017*; *Kurokawa et al., 2022*). Nine evolutionary lineages were all initiated from the identical culture stock prepared in advance. The 24-well microplates specific for microbe culture (IWAKI) were used, and every four wells of four tenfold serial dilutions, for example, $10^3$–$10^6$, were used for each lineage. Multiple dilutions changing in order promised at least one of the wells within the exponential growth phase after the overnight culture. The microplates were incubated in a microplate bioshaker (Deep Well Maximizer, Taitec) at 37°C, with rotation at 500 rpm. The serial transfer was performed at ~24 hr intervals. Only one of the four wells (dilutions) showing growth in the early exponential phase ($OD_{600}$ = 0.01–0.1) was selected and diluted into four wells of a new microplate using four dilution ratios. Serial transfer was repeated until all evolutionary lineages reached approximately 1000 generations, which required approximately 2 mo per lineage. A total of nine lineages were conducted, and the daily records of the nine lineages are summarized in *Supplementary file 1*. The evolutionary generations (*G*) and the growth rates (*µ*) were calculated according to *Equations 1 and 2*.

$$G = \log_2\left(\frac{N_t}{N_0}\right) \tag{1}$$

$$\mu = \frac{\ln\left(\frac{N_t}{N_0}\right)}{\Delta t} \tag{2}$$

## Growth assay

Both the ancestor and the evolved *E. coli* populations of the reduced genome were subjected to the growth assay, as previously described (*Kurokawa et al., 2016*; *Kurokawa and Ying, 2017*). In brief, every 200 µl of culture was dispensed to each well in the 96-well microplate (Coaster). The microplate was incubated at 37°C in a plate reader (EPOCH2, BioTek), shaking at 567 cpm (cycles per minute) for 48 hr. The temporal changes in $OD_{600}$ were measured at 30 min intervals. The growth fitness (*r*) was calculated using the following equation between any two consecutive points (*Equation 3*).

$$r_i = \frac{\ln\left(\frac{C_{i+1}}{C_i}\right)}{t_{i+1} - t_i} \tag{3}$$

where $t_i$ and $t_{i+1}$ are the culture times at the two consecutive measurement points, and $C_i$ and $C_{i+1}$ are the $OD_{600}$ at time points $t_i$ and $t_{i+1}$. The growth rate was determined as the average of three consecutive $r_i$, showing the largest mean and minor variance to avoid the unreliable calculation caused by the occasionally occurring values. The details of the growth assay operation can be found at https://doi.org/10.3791/56197. The mean of the biological triplicates was defined as the growth fitness and used in the analyses.

## Genome resequencing and mutation analysis

The *E. coli* cells were collected at the stationary growth phase (i.e., $OD_{600} > 1.0$) and subjected to genome resequencing, as described previously (*Kurokawa et al., 2022*). In brief, the bacterial culture was stopped by adding rifampicin at 300 µg/ml. The cell pellet was collected for genomic DNA purification using a Wizard Genomic DNA Purification Kit (Promega) under the manufacturer's instructions. The sequencing library was prepared using the Nextera XT DNA Sample Prep Kit (Illumina), and the paired-end sequencing (300 bp ×2) was performed using the Illumina HiSeq platform. The raw datasets of DNA-seq were deposited in the DDBJ Sequence Read Archive under the accession number DRA013661. The sequencing reads were aligned to the reference genome *E. coli* W3110 (AP009048.1, GenBank), and the mutation analysis was performed with the Breseq pipeline (*Deatherage et al., 2014*; *Barrick et al., 2014*). The DNA sequencing and mapping statistics are summarized in *Supplementary file 2*. The mutations fixed in the coding regions due to evolution are shown in *Supplementary file 3*.

## Calculation and simulation of the genomic positions of mutations

The distances from the genomic positions of the genome mutations fixed in the nine Evos to the nearest genomic scars caused by the genome reduction were calculated as previously described *Kurokawa et al., 2016*; *Ying et al., 2013*. The mutation simulation was performed with Python in the following steps. A total of 65 mutations were randomly generated on the reduced genome, and the distances from the mutated genomic locations to the nearest genomic scars caused by genome reduction were calculated. Subsequently, Welch's *t*-test was performed to evaluate whether the distances calculated from the random mutations were significantly longer or shorter than those calculated from the mutations that occurred in Evos. The random simulation, distance calculation, and statistic test were performed 1000 times, which resulted in 1000 p-values. Finally, the mean of p-values ($\mu_p$) was calculated, and a 95% reliable region was applied. It was used to evaluate whether the 65 mutations in the Evos were significantly close to the genomic scars, that is, the locational bias.

## RNA sequencing

The *E. coli* cells were collected at the exponential growth phase (i.e., $5 \times 10^7 - 2 \times 10^8$ cells/ml) and subjected to RNA sequencing, as described previously (*Matsui et al., 2023*; *Liu et al., 2020*). In brief, the bacterial growth was stopped by mixing with the iced 10% phenol ethanol solution. The cell pellet was collected to purify the total RNAs using the RNeasy Mini Kit (QIAGEN) and RNase-Free DNase Set (QIAGEN) according to product instructions. The paired-end sequencing (150 bp ×2) was performed using the Novaseq6000 next-generation sequencer (Illumina). The rRNAs were removed from the total RNAs using the Ribo-Zero Plus rRNA Depletion Kit (Illumina), and the mRNA libraries were prepared using the Ultra Directional RNA Library Prep Kit for Illumina (NEBNext). Biological replicates were performed for all conditions (N = 2–4). The raw datasets were deposited in the DDBJ Sequence Read Archive under the accession number DRA013662.

## Data processing and normalization

The FASTQ files were mapped to the reference genome W3110 (accession number AP009048.1, GenBank) using the mapping software Bowtie2 (*Langmead and Salzberg, 2012*), as described previously (*Matsui et al., 2023*; *Liu et al., 2020*). The obtained read counts were converted to FPKM values according to the gene length and total read count values. Global normalization of the FPKM values was performed to reach an identical mean value in the logarithmic scale in all datasets. The resulting normalized FPKM values were statistically equivalent to TPM. The gene expression level was determined as the logarithmic value of FPKM, and the mean values of biological replicates were used for the following analyses (*Supplementary file 7*).

## Computational analysis

The normalized datasets were subjected to the computational analyses performed with the R statistical analysis software. A total of 3290 genes were used for hierarchical clustering and PCA, which were performed with the R functions of 'hclust' and 'prcomp', respectively, as described previously (*Matsui et al., 2023*). The corresponding parameters of 'method' and 'scale' were set as 'average' and 'F', respectively. The R package of DESeq2 (*Love et al., 2014*) was used to determine the DEGs, based on the false discovery rate (FDR < 0.05) (*Storey, 2002*). The read counts were used as the input data for DESeq2, in which the data normalization was performed at each run for pair comparison.

## Enrichment analysis

Functional enrichments were performed according to the features of gene category (*Riley et al., 2006*), transcriptional regulation (*Salgado et al., 2013*), GO (*Ashburner et al., 2000*), and metabolic pathways (*Kanehisa et al., 2016*; *Kanehisa and Goto, 2000*). In total, 21 gene categories and 46 regulons, which comprised more than 15 genes and 15 regulatees, were subjected to the enrichment, respectively. The statistical significance was evaluated using the binomial test with Bonferroni correction. The enrichment analysis of GO (GO terms) (*Ashburner et al., 2000*; *Carbon et al., 2009*) and metabolic pathway (KEGG) (*Kanehisa et al., 2016*; *Kanehisa and Goto, 2000*) was performed using DAVID, a web-based tool for visualizing the characteristics of gene clusters with expression variation (*Huang et al., 2007*; *Huang et al., 2009*). The statistical significance was according to FDR.

## Chromosomal periodicity analysis

Fourier transform was used to evaluate the chromosomal periodicity of the transcriptome, as previously described (*Nagai et al., 2020*; *Liu et al., 2020*). The genome was divided into compartments of 1 kb each, and the mean expression level of the genes within the corresponding sections was calculated. Gene expression levels were smoothed with a moving average of 100 kb and subjected to the periodicity analysis using the function 'periodogram' in R. The max peak (periodic wavelength) of the periodogram was fitted to the gene expression data using the function 'nls' in R by the least-squares method according to *Equation 4*.

$$exp\left(x\right) = a * sin\left(x + \frac{b}{T}\right) + c \tag{4}$$

where *a, b, T,* and *c* represent the periodic amplitude, periodic phase, periodic wavelength indicated by the max peak, and mean expression level of the whole transcriptome as a constant, respectively. Note that altering the moving average did not change the max peak. The statistical significance of the periodicity was assessed with Fisher's g test (*Supplementary file 5*), as described previously (*Nagai et al., 2020*; *Liu et al., 2020*). The genomic position of *ori* was determined according to the previous reports (*Liu et al., 2020*; *Bryant et al., 2014*). In addition, the function 'abline' in R was used to point out the genomic positions of the mutations.

## Gene network analysis

The WGCNA of the nine Evos was performed with the R package of WGCNA (*Langfelder and Horvath, 2008*), as described previously (*Matsui et al., 2023*). A step-by-step method was used to determine the parameters for constructing the gene networks. The soft threshold was set at 12, where the $R^2$ of Scale Free Topology Model Fit was approximately 0.9, as recommended by the developer's instruction. The resultant gene networks were clustered with the 'hclust' function (method = average)

and reconstructed by merging similar modules using the 'mergeCloseModules' function with a height cut of 0.25 in the 'dynamic tree cut' method. Finally, a total of 21 modules were determined for 3290 genes. The correlation coefficients and p-values between the expression of the gene modules and the other global features (growth rates, DEGs, and mutations) were evaluated using the functions 'cor' and 'corPvalueStudent' in R, respectively. FDR correction was applied to the p-values to account for multiplicity. Functional enrichment of the gene modules was performed, and the statistical significance was evaluated by the binomial test with Bonferroni correction as described above.

## Acknowledgements

We thank NBRP for providing the *E. coli* strains carrying the wild-type and reduced genomes (W3110 and KHK collection). This work was supported by the JSPS KAKENHI Grant-in-Aid for Scientific Research (B) (grant number 19H03215) and partially by Grant-in-Aid for Challenging Exploratory Research (grant number 21K19815).

## Additional information

### Funding

| Funder | Grant reference number | Author |
|---|---|---|
| Japan Society for the Promotion of Science | 19H03215 | Bei-Wen Ying |
| Japan Society for the Promotion of Science | 21K19815 | Bei-Wen Ying |

The funders had no role in study design, data collection and interpretation, or the decision to submit the work for publication.

### Author contributions

Kenya Hitomi, Data curation, Formal analysis, Visualization, Methodology, Writing - original draft; Yoichiro Ishii, Resources, Methodology; Bei-Wen Ying, Conceptualization, Resources, Formal analysis, Supervision, Funding acquisition, Validation, Investigation, Visualization, Writing - original draft, Project administration, Writing - review and editing

### Author ORCIDs

Bei-Wen Ying http://orcid.org/0000-0003-2517-5686

Reviewer #1 (Public Review): https://doi.org/10.7554/eLife.93520.3.sa1
Reviewer #2 (Public Review): https://doi.org/10.7554/eLife.93520.3.sa2
Reviewer #3 (Public Review): https://doi.org/10.7554/eLife.93520.3.sa3
Author response https://doi.org/10.7554/eLife.93520.3.sa4

## Additional files

### Supplementary files

- Supplementary file 1. Daily records of experimental evolution. The time and OD$_{600}$ of overnight culture, the well (dilution rate) used for the serial transfer, and the calculated generation and growth rate are shown. All nine evolutionary lineages are summarized.

- Supplementary file 2. Statistics of genome sequencing. The parameters acquired in the genome resequencing, which represent the goodness of the sequencing, are summarized.

- Supplementary file 3. List of genome mutations. The total 65 mutations fixed in the Evos are summarized. The type of mutation, the position in the reduced genome, distance to the nearest genomic scar, changes in DNA and amino acid, gene function, essentiality (e, essential; n, nonessential), gene category, etc., are indicated. Note that the entire population held the mutations, i.e., 100% frequency in DNA sequencing.

• Supplementary file 4. Gene categories comprising the mutated genes. The full names and abbreviations of the gene categories are shown. The number of mutated genes in each gene category is tallied for each Evos.

• Supplementary file 5. Statistics of chromosomal periodicity of transcriptomes. The maximal peak (wavelength) acquired by the Fourier transform, the number of periods resulting from the curve fitting, the Fisher's g test, and the p-values are summarized.

• Supplementary file 6. List of the overlapped DEGs. A total of 108 DEGs are summarized. Gene ID, gene name, and gene function are indicated.

• Supplementary file 7. Datasheet of normalized gene expression. Gene expression levels are shown in the logarithmic value of FKPM. Gene ID, gene name, strain name, and nine evolutionary lineages are indicated. N0 and N28 represent the wild-type and reduced genomes, respectively.

• MDAR checklist

## Data availability

All data generated or analyzed during this study are included in the manuscript and supporting files. DNA and RNA sequencing data sets have been deposited in the DDBJ Sequence Read Archive under the accession numbers DRA013661 and DRA013662, respectively.

The following datasets were generated:

| Author(s) | Year | Dataset title | Dataset URL | Database and Identifier |
| --- | --- | --- | --- | --- |
| Ishii Y, Ying BW | 2023 | Elucidating patterns of adaptive evolution in genome-reduced *Escherichia coli* | https://ddbj.nig.ac.jp/resource/sra-submission/DRA013661 | DDBJ Sequence Read Archive, DRA013661 |
| Ying BW, Ishii Y | 2023 | Elucidating patterns of adaptive evolution in genome-reduced *Escherichia coli* | https://ddbj.nig.ac.jp/resource/sra-submission/DRA013662 | DDBJ Sequence Read Archive, DRA013662 |

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
