## [Editor Report · eLife assessment]

This is an **important** study of the recovery of genome-reduced bacterial cells in laboratory evolution experiments to understand how they regain their fitness. Through the analysis of gene expression and a series of tests, the authors present **convincing** evidence indicating distinct molecular changes in the evolved bacterial strains, although the precise mechanisms remain uncharacterized. These findings imply that diverse mechanisms are employed to offset the effects of a reduced genome, offering intriguing insights into genome evolution.

---

## [Referee Report · Reviewer #1 (Public Review)]

In this study, the authors explored how the reduced growth fitness, resulting from genome reduction, can be compensated through evolution. They conducted an evolution experiment with a strain of *Escherichia coli* that carried a reduced genome, over approximately 1,000 generations. The authors carried out sequencing, and found no clear genetic signatures of evolution across replicate populations. They carry out transcriptomics and a series of analyses that lead them to conclude that there are divergent mechanisms at play in individual evolutionary lineages. The authors used gene network reconstruction to identify three gene modules functionally differentiated, correlating with changes in growth fitness, genome mutation, and gene expression, respectively, due to evolutionary changes in the reduced genome.

I think that this study addresses an interesting question. Many microbial evolution experiments evolve by loss of function mutations, but presumably a cell that has already lost so much of its genome needs to find other mechanisms to adapt. Experiments like this have the potential to study "constructive" rather than "destructive" evolution.

Comments on revised version:

I think the authors have carefully gone through the manuscript and addressed all of my concerns.

---

## [Referee Report · Reviewer #2 (Public Review)]

This manuscript describes an adaptive laboratory evolution (ALE) study with a previously constructed genome-reduced *E. coli*. The growth performance of the end-point lineages evolved in M63 medium was comparable to the full-length wild-type level at lower cell densities.

Subsequent mutation profiling and RNA-Seq analysis revealed many changes on the genome and transcriptomes of the evolved lineages. The authors did a great deal on analyzing the patterns of evolutionary changes between independent lineages, such as the chromosomal periodicity of transcriptomes, pathway enrichment analysis, weight gene co-expression analysis, and so on. They observed a striking diversity in the molecular characteristics amongst the evolved lineages, which, as they suggest, reflect divergent evolutionary strategies adopted by the genome-reduced organism.

As for the overall quality of the manuscript, I am rather torn. The manuscript leans towards elaborating observed findings, rather than explaining their biological significance. For this reason, readers are left with more questions than answers. For example, fitness assay on reconstituted (single and combinatorial) mutants was not performed, nor any supporting evidence on the proposed contributions of each mutants provided. This leaves the nature of mutations - be them beneficial, neutral or deleterious, the presence of epistatic interactions, and the magnitude of fitness contribution, largely elusive. Also, it is difficult to tell whether the RNA-Seq analysis in this study managed to draw biologically meaningful conclusions, or instill insight into the nature of genome-reduced bacteria. The analysis primarily highlighted the differences in transcriptome profiles among each lineage based on metrics such as 'DEG counts' and the 'GO enrichment'. However, I could not see any specific implications regarding the biology of the evolved minimal genome drawn. In their concluding remark, 'Multiple evolutionary paths for the reduced genome to improve growth fitness were likely all roads leading to Rome,' the authors observed the first half of the sentence, but the distinctive characteristics of 'all roads' or 'evolutionary paths', which I think should have been the key aspect in this investigation, remains elusive.

Comments on revised version:

I appreciate the author's responses. They responded to most of the comments, but I still think that there is room for improvement. Please refer to the following comments. Quoted below are the author's responses.

"We agree that our study leaned towards elaborating observed findings rather than explaining the detailed biological mechanisms."

- Comment: I doubt if there are scientific merits in merely elaborating observed findings. The conclusion of this study suggests that evolutionary paths in reduced genomes are highly diverse. But if you think about the nature of adaptive evolution, which relies upon the spontaneous mutation event followed by selection, certain degree of divergence is always expected. The problem with current experimental setting is that there are no ways to quantitively assess whether the degree of evolutionary divergence increases as the function of genome reduction, as the authors claimed. In addition, this notion is in direct contradiction to the prediction that genome reduction constraints evolution by reducing the number of solution space. It is more logical to think and predict that genome reduction would, in turn, lead to the loss of evolutionary divergence. We are also interested to know whether solution space to the optimization problem altered in response to the genome reduction. In this regard, a control ALE experiment on non-reduced wild-type seems to be a mandatory experimental control. I highly suggest that authors present a control experiment, as it was done for "JCVI syn3.0B vs non-minimal M. mycoides" (doi: 10.1038/s41586 023 06288 x) and "*E. coli* eMS57 vs MG1655" (doi: 10.1038/s41467 019 08888 6).

"We focused on the genome wide biological features rather than the specific biological functions."

- Comment: The 'biological features' delivered in current manuscript does not give insight as to which genomic changes translated into strain fitness improvement. Rather than explaining the genotype-phenotype relationships and/or the mechanistic basis of fitness improvement, authors merely elaborated on the observed phenotypes. I question the scientific merits of such 'findings'.

"Although the reduced growth rate caused by genome reduction could be recovered by experimental evolution, it remains unclear whether such an evolutionary improvement in growth fitness was a general feature of the reduced genome and how the genome wide changes occurred to match the growth fitness increase."

- Comment: This response is very confusing to understand. "it remains unclear whether such an evolutionary improvement in growth fitness was a general feature of the reduced genome" - what aspects remain unclear?? What assumption led the authors to believe that reduced genome's fitness cannot be evolutionarily improved?

- Comment: "and how the genome wide changes occurred to match the growth fitness increase" - this is exactly the aspect that authors should deliver, instead of just elaborating the observed findings. Why don't authors select one or two fastest-growing (or the fittest) lineages and specifically analyze underlying adaptive changes (i.e. genotype-phenotype relationships)?

---

## [Referee Report · Reviewer #3 (Public Review)]

Summary:

Studying evolutionary trajectories provides important insight in genetic architecture of adaptation and provide potential contribution to evaluating the predictability (or unpredictability) in biological processes involving adaptation. While many papers in the field address adaptation to environmental challenges, the number of studies on how genomic contexts, such as large-scale variation, can impact evolutionary outcomes adaptation is relatively low. This research experimentally evolved a genome-reduced strain for ~1000 generations with 9 replicates and dissected their evolutionary changes. Using the fitness assay of OD measurement, the authors claimed there is a general trend of increasing growth rate and decreasing carrying capacity, despite a positive correlation among all replicates. The authors also performed genomic and transcriptomic research at the end of experimental evolution, claiming the dissimilarity in the evolution at the molecular level.

Strengths:

The experimental evolution approach with a high number of replicates provides a good way to reveal the generality/diversity of the evolutionary routes.

The assay of fitness, genome, and transcriptome all together allows a more thorough understanding of the evolutionary scenarios and genetic mechanisms.

Comments on revised version:

#5 in the last round of comments: When the authors mentioned no overlapping in single mutation level, I thought the authors would directly use this statement to support their next sentence about no bias of these mutations. As the author's responded, I was suspecting no overlapping for 65 mutation across the entire genome is likely to be not statistically significant. In the revised version, the authors emphasized and specified their simulation and argument in the following sentences, so I do not have questions on this point anymore.

#14 in the last round of comments: As what authors responded, "short-term responses" meant transcriptional or physiological changes within a few hours after environmental or genetic fluctuation. "long-term responses" involve new compensatory mutations and selection. The point was that, the authors found that "the transcriptome reorganization for fitness increase triggered by evolution differed from that for fitness decrease caused by genome reduction." That is short vs long-term responses to genetic perturbation. Some other experimental evolution did short vs long-term responses to environmental perturbation and usually also found that the short-term responses are reverted in the long-term responses (e.g., https://academic.oup.com/mbe/article/33/1/25/2579742). I hope this explanation makes more sense. And I think the authors can make their own decisions on whether they would like to add this discussion or not.

---

## [Author Response]

The following is the authors’ response to the original reviews.

**Response to Reviewer #1:**

Thank you for the careful reading and the positive evaluation of our manuscript. As you mentioned, the present study tried to address the question of how the lost genomic functions could be compensated by evolutionary adaptation, indicating the potential mechanism of "constructive" rather than "destructive" evolution. Thank you for the instructive comments that helped us to improve the manuscript. We sincerely hope the revised manuscript and the following point-to-point response meet your concerns.

Line 80 "Growth Fitness" is this growth rate?

Yes. The sentence was revised as follows.

(L87-88) “The results demonstrated that most evolved populations (Evos) showed improved growth rates, in which eight out of nine Evos were highly significant (Fig. 1B, upper).”

Line 94 a more nuanced understanding of r/K selection theory, allows for trade-ups between R and K, as well as trade-offs. This may explain why you did not see a trade-off between growth and carrying capacity in this study. See this paper https://doi.org/10.1038/s41396-023-01543-5. Overall, your evos lineages evolved higher growth rates and lower carrying capacity (Figures 1B, C, E). If selection was driving the evolution of higher growth rates, it may have been that there was no selective pressure to maintain high carrying capacity. This means that the evolutionary change you observed in carrying capacity may have been neutral "drift" of the carrying capacity trait, during selection for growth rate, not because of a trade-off between R and K. This is especially likely since carrying capacity declined during evolution. Unless the authors have convincing evidence for a tradeoff, I suggest they remove this claim.Line 96 the authors introduce a previous result where they use colony size to measure growth rate, this finding needs to be properly introduced and explained so that we can understand the context of the conclusion.Line 97 This sentence "the collapse of the trade-off law likely resulted from genome reduction." I am not sure how the authors can draw this conclusion, what is the evidence supporting that the genome size reduction causes the breakdown of the tradeoff between R and K (if there was a tradeoff)?

Thank you for the reference information and the thoughtful comments. The recommended paper was newly cited, and the description of the trade-off collapse was deleted. Accordingly, the corresponding paragraph was rewritten as follows.

(L100-115) “Intriguingly, a positive correlation was observed between the growth fitness and the carrying capacity of the Evos (Fig. 1D). It was somehow consistent with the positive correlations between the colony growth rate and the colony size of a genome-reduced strain 11 and between the growth rates and the saturated population size of an assortment of genome reduced strains 13. Nevertheless, the negative correlation between growth rate and carrying capacity, known as the r/K selection30,31 was often observed as the trade-off relationship between r and K in the evolution and ecology studies 32 33,34. As the r/K trade-off was proposed to balance the cellular metabolism that resulted from the cost of enzymes involved 34, the deleted genes might play a role in maintaining the metabolism balance for the r/K correlation. On the other hand, the experimental evolution (i.e., serial transfer) was strictly performed within the exponential growth phase; thus, the evolutionary selection was supposed to be driven by the growth rate without selective pressure to maintain the carrying capacity. The declined carrying capacity might have been its neutral "drift" but not a trade-off to the growth rate. Independent and parallel experimental evolution of the reduced genomes selecting either r or K is required to clarify the actual mechanisms.”

Line 103 Genome mutations. The authors claim that there are no mutations in parallel but I see that there is a 1199 base pair deletion in eight of the nine evo strains (Table S3). I would like the author to mention this and I'm actually curious about why the authors don't consider this parallel evolution.

Thank you for your careful reading. According to your comment, we added a brief description of the 1199-bp deletion detected in the Evos as follows.

(L119-122) “The number of mutations largely varied among the nine Evos, from two to 13, and no common mutation was detected in all nine Evos (Table S3). A 1,199-bp deletion of insH was frequently found in the Evos (Table S3, highlighted), which well agreed with its function as a transposable sequence.”

Line 297 Please describe the media in full here - this is an important detail for the evolution experiment. Very frustrating to go to reference 13 and find another reference, but no details of the method. Looked online for the M63 growth media and the carbon source is not specified. This is critical for working out what selection pressures might have driven the genetic and transcriptional changes that you have measured. For example, the parallel genetic change in 8/9 populations is a deletion of insH and tdcD (according to Table S3). This is acetate kinase, essential for the final step in the overflow metabolism of glucose into acetate. If you have a very low glucose concentration, then it could be that there was selection to avoid fermentation and devote all the pyruvate that results from glycolysis into the TCA cycle (which is more efficient than fermentation in terms of ATP produced per pyruvate).

Sorry for the missing information on the medium composition, which was additionally described in the Materials and Methods. The glucose concentration in M63 was 22 mM, which was supposed to be enough for bacterial growth. Thank you for your intriguing thinking about linking the medium component to the genome mutation-mediated metabolic changes. As there was no experimental result regarding the biological function of gene mutation in the present study, please allow us to address this issue in our future work.

(L334-337) “In brief, the medium contains 62 mM dipotassium hydrogen phosphate, 39 mM potassium dihydrogen phosphate, 15 mM ammonium sulfate, 15 μM thiamine hydrochloride, 1.8 μM Iron (II) sulfate, 0.2 mM magnesium sulfate, and 22 mM glucose.”

Line 115. I do not understand this argument "They seemed highly related to essentiality, as 11 out of 49 mutated genes were essential (Table S3)." Is this a significant enrichment compared to the expectation, i.e. the number of essential genes in the genome? This enrichment needs to be tested with a Hypergeometric test or something similar.Also, "As the essential genes were known to be more conserved than nonessential ones, the high frequency of the mutations fixed in the essential genes suggested the mutation in essentiality for fitness increase was the evolutionary strategy for reduced genome." I do not think that there is enough evidence to support this claim, and it should be removed.

Sorry for the unclear description. Yes, the mutations were significantly enriched in the essential genes (11 out of 45 genes) compared to the essential genes in the whole genome (286 out of 3290 genes). The improper description linking the mutation in essential genes to the fitness increase was removed, and an additional explanation on the ratio of essential genes was newly supplied as follows.

(L139-143) “The ratio of essential genes in the mutated genes was significantly higher than in the total genes (286 out of 3290 genes, Chi-square test p=0.008). As the essential genes were determined according to the growth35 and were known to be more conserved than nonessential ones 36,37, the high frequency of the mutations fixed in the essential genes was highly intriguing and reasonable.”

Line 124 Regarding the mutation simulations, I do not understand how the observed data were compared to the simulated data, and how conclusions were drawn. Can the authors please explain the motivation for carrying out this analysis, and clearly explain the conclusions?

Random simulation was additionally explained in the Materials and Methods and the conclusion of the random simulation was revised in the Results, as follows.

(L392-401) “The mutation simulation was performed with Python in the following steps. A total of 65 mutations were randomly generated on the reduced genome, and the distances from the mutated genomic locations to the nearest genomic scars caused by genome reduction were calculated. Subsequently, Welch's t-test was performed to evaluate whether the distances calculated from the random mutations were significantly longer or shorter than those calculated from the mutations that occurred in Evos. The random simulation, distance calculation, and statistic test were performed 1,000 times, which resulted in 1,000 p values. Finally, the mean of p values (μp) was calculated, and a 95% reliable region was applied. It was used to evaluate whether the 65 mutations in the Evos were significantly close to the genomic scars, i.e., the locational bias.”

(L148-157) “Random simulation was performed to verify whether there was any bias or hotspot in the genomic location for mutation accumulation due to the genome reduction. A total of 65 mutations were randomly generated on the reduced genome (Fig. 2B), and the genomic distances from the mutations to the nearest genome reduction-mediated scars were calculated. Welch's t-test was performed to evaluate whether the genomic distances calculated from random mutations significantly differed from those from the mutations accumulated in the Evos. As the mean of p values (1,000 times of random simulations) was insignificant (Fig. 2C, μp > 0.05), the mutations fixed on the reduced genome were either closer or farther to the genomic scars, indicating there was no locational bias for mutation accumulation caused by genome reduction.”

Line 140 The authors should give some background here - explain the idea underlying chromosomal periodicity of the transcriptome, to help the reader understand this analysis.Line 142 Here and elsewhere, when referring to a method, do not just give the citation, but also refer to the methods section or relevant supplementary material.

The analytical process (references and methods) was described in the Materials and Methods, and the reason we performed the chromosomal periodicity was added in the Results as follows.

(L165-172) “As the *E. coli* chromosome was structured, whether the genome reduction caused the changes in its architecture, which led to the differentiated transcriptome reorganization in the Evos, was investigated. The chromosomal periodicity of gene expression was analyzed to determine the structural feature of genome-wide pattern, as previously described 28,38. The analytical results showed that the transcriptomes of all Evos presented a common six-period with statistical significance, equivalent to those of the wild-type and ancestral reduced genomes (Fig. 3A, Table S4).”

Line 151 "The expression levels of the mutated genes were higher than those of the remaining genes (Figure 3B)"- did this depend on the type of mutation? There were quite a few early stops in genes, were these also more likely to be expressed? And how about the transcriptional regulators, can you see evidence of their downstream impact?

Sorry, we didn't investigate the detailed regulatory mechanisms of 49 mutated genes, which was supposed to be out of the scope of the present study. Fig. 3B was the statistical comparison between 3225 and 49 genes. It didn't mean that all mutated genes expressed higher than the others. The following sentences were added to address your concern.

(L181-185) “As the regulatory mechanisms or the gene functions were supposed to be disturbed by the mutations, the expression levels of individual genes might have been either up- or down-regulated. Nevertheless, the overall expression levels of all mutated genes tended to be increased. One of the reasons was assumed to be the mutation essentiality, which remained to be experimentally verified.”

Line 199 onward. The authors used WGCNA to analyze the gene expression data of evolved organisms. They identified distinct gene modules in the reduced genome, and through further analysis, they found that specific modules were strongly associated with key biological traits like growth fitness, gene expression changes, and mutation rates. Did the authors expect that there was variation in mutation rate across their populations? Is variation from 3-16 mutations that they observed beyond the expectation for the wt mutation rate? The genetic causes of mutation rate variation are well understood, but I could not see any dinB, mutT,Y, rad, or pol genes among the discovered mutations. I would like the authors to justify the claim that there was mutation rate variation in the evolved populations.

Thank you for the intriguing thinking. We don't think the mutation rates were significantly varied across the nine populations, as no mutation occurred in the MMR genes, as you noticed. Our previous study showed that the spontaneous mutation rate of the reduced genome was higher than that of the wild-type genome (Nishimura et al., 2017, mBio). As nonsynonymous mutations were not detected in all nine Evos, the spontaneous mutation rate couldn't be calculated (because it should be evaluated according to the ratio of nonsynonymous and synonymous single-nucleotide substitutions in molecular evolution). Therefore, discussing the mutation rate in the present study was unavailable. The following sentence was added for a better understanding of the gene modules.

(L242-245) “These modules M2, M10 and M16 might be considered as the hotspots for the genes responsible for growth fitness, transcriptional reorganization, and mutation accumulation of the reduced genome in evolution, respectively.”

Line 254 I get the idea of all roads leading to Rome, which is very fitting. However, describing the various evolutionary strategies and homeostatic and variable consequence does not sound correct - although I am not sure exactly what is meant here. Looking at Figure 7, I will call strategy I "parallel evolution", that is following the same or similar genetic pathways to adaptation and strategy ii I would call divergent evolution. I am not sure what strategy iii is. I don't want the authors to use the terms parallel and divergent if that's not what they mean. My request here would be that the authors clearly describe these strategies, but then show how their results fit in with the results, and if possible, fit with the naming conventions, of evolutionary biology.

Thank you for your kind consideration and excellent suggestion. It's our pleasure to adopt your idea in tour study. The evolutionary strategies were renamed according to your recommendation. Both the main text and Fig. 7 were revised as follows.

(L285-293) “Common mutations22,44 or identical genetic functions45 were reported in the experimental evolution with different reduced genomes, commonly known as parallel evolution (Fig. 7, i). In addition, as not all mutations contribute to the evolved fitness 22,45, another strategy for varied phenotypes was known as divergent evolution (Fig. 7, ii). The present study accentuated the variety of mutations fixed during evolution. Considering the high essentiality of the mutated genes (Table S3), most or all mutations were assumed to benefit the fitness increase, partially demonstrated previously 20. Nevertheless, the evolved transcriptomes presented a homeostatic architecture, revealing the divergent to convergent evolutionary strategy (Fig. 7, iii).”

**Author response image 1. sa4fig1:** 

Line 327 Growth rates/fitness. I don't think this should be called growth fitness- a rate is being calculated. I would like the authors to explain how the times were chosen - do the three points have to be during the log phase? Can you also explain what you mean by choosing three ri that have the largest mean and minor variance?

Sorry for the confusing term usage. The fitness assay was changed to the growth assay. Choosing three ri that have the largest mean and minor variance was to avoid the occasional large values (blue circle), as shown in the following figure. In addition, the details of the growth analysis can be found at https://doi.org/10.3791/56197 (ref. 59), where the video of experimental manipulation, protocol, and data analysis is deposited. The following sentence was added in accordance.

**Author response image 2. sa4fig2:** 

(L369-371) “The growth rate was determined as the average of three consecutive ri, showing the largest mean and minor variance to avoid the unreliable calculation caused by the occasionally occurring values. The details of the experimental and analytical processes can be found at https://doi.org/10.3791/56197.”

Line 403 Chromosomal periodicity analysis. The windows chosen for smoothing (100kb) seem big. Large windows make sense for some things - for example looking at how transcription relates to DNA replication timing, which is a whole-genome scale trend. However, here the authors are looking for the differences after evolution, which will be local trends dependent on specific genes and transcription factors. 100kb of the genome would carry on the order of one hundred genes and might be too coarse-grained to see differences between evos lineages.

Thank you for the advice. We agree that the present analysis focused on the global trend of gene expression. Varying the sizes may lead to different patterns. Additional analysis was performed according to your comment. The results showed that changes in window size (1, 10, 50, 100, and 200 kb) didn't alter the periodicity of the reduced genome, which agreed with the previous study on a different reduced genome MDS42 of a conserved periodicity (Ying et al., 2013, BMC Genomics). The following sentence was added in the Materials and Methods.

(L460-461) “Note that altering the moving average did not change the max peak.”

Figures - the figures look great. Figure 7 needs a legend.

Thank you. The following legend was added.

(L774-777) “Three evolutionary strategies are proposed. Pink and blue arrowed lines indicate experimental evolution and genome reduction, respectively. The size of the open cycles represents the genome size. Black and grey indicate the ancestor and evolved genomes, respectively.”

**Response to Reviewer #2:**

Thank you for reviewing our manuscript and for your fruitful comments. We agree that our study leaned towards elaborating observed findings rather than explaining the detailed biological mechanisms. We focused on the genome-wide biological features rather than the specific biological functions. The underlying mechanisms indeed remained unknown, leaving the questions as you commented. We didn't perform the fitness assay on reconstituted (single and combinatorial) mutants because the research purpose was not to clarify the regulatory or metabolic mechanisms. It's why the RNA-Seq analysis provided the findings on genome-wide patterns and chromosomal view, which were supposed to be biologically valuable. We did understand your comments and complaints that the conclusions were biologically meaningless, as ALE studies that found the specific gene regulation or improved pathway was the preferred story in common, which was not the flow of the present study.

For this reason, our revision may not address all these concerns. Considering your comments, we tried our best to revise the manuscript. The changes made were highlighted. We sincerely hope the revision and the following point-to-point response are acceptable.

Major remarks:(1) The authors outlined the significance of ALE in genome-reduced organisms and important findings from published literature throughout the Introduction section. The description in L65-69, which I believe pertains to the motivation of this study, seems vague and insufficient to convey the novelty or necessity of this study i.e. it is difficult to grasp what aspects of genome-reduced biology that this manuscript intends to focus/find/address.

Sorry for the unclear writing. The sentences were rewritten for clarity as follows.

(L64-70) “Although the reduced growth rate caused by genome reduction could be recovered by experimental evolution, it remains unclear whether such an evolutionary improvement in growth fitness was a general feature of the reduced genome and how the genome-wide changes occurred to match the growth fitness increase. In the present study, we performed the experimental evolution with a reduced genome in multiple lineages and analyzed the evolutionary changes of the genome and transcriptome.”

(2) What is the rationale behind the lineage selection described in Figure S1 legend "Only one of the four overnight cultures in the exponential growth phase (OD600 = 0.01~0.1) was chosen for the following serial transfer, highlighted in red."?

The four wells (cultures of different initial cell concentrations) were measured every day, and only the well that showed OD600=0.01~0.1 (red) was transferred with four different dilution rates (e.g., 10, 100, 1000, and 10000 dilution rates). It resulted in four wells of different initial cell concentrations. Multiple dilutions promised that at least one of the wells would show the OD600 within the range of 0.01 to 0.1 after the overnight culture. They were then used for the next serial transfer. Fig. S1 provides the details of the experimental records. The experimental evolution was strictly controlled within the exponential phase, quite different from the commonly conducted ALE that transferred a single culture in a fixed dilution rate. Serial transfer with multiple dilution rates was previously applied in our evolution experiments and well described in Nishimura et al., 2017, mBio; Lu et al., 2022, Comm Biol; Kurokawa et al., 2022, Front Microbiol, etc. The following sentence was added in the Materials and Methods.

(L344-345) “Multiple dilutions changing in order promised at least one of the wells within the exponential growth phase after the overnight culture.”

(3) The measured growth rate of the end-point 'F2 lineage' shown in Figure S2 seemed comparable to the rest of the lineages (A1 to H2), but the growth rate of 'F2' illustrated in Figure 1B indicates otherwise (L83-84). What is the reason for the incongruence between the two datasets?

Sorry for the unclear description. The growth rates shown in Fig. S2 were obtained during the evolution experiment using the daily transfer's initial and final OD600 values. The growth rates shown in Fig. 1B were obtained from the final population (Evos) growth assay and calculated from the growth curves (biological replication, N=4). Fig. 1B shows the precisely evaluated growth rates, and Fig. S2 shows the evolutionary changes in growth rates. Accordingly, the following sentence was added to the Results.

(L84-87) “As the growth increases were calculated according to the initial and final records, the exponential growth rates of the ancestor and evolved populations were obtained according to the growth curves for a precise evaluation of the evolutionary changes in growth.”

(4) Are the differences in growth rate statistically significant in Figure 1B?

Eight out of nine Evos were significant, except F2. The sentences were rewritten and associated with the revised Fig. 1B, indicating significance.

(L87-90) “The results demonstrated that most evolved populations (Evos) showed improved growth rates, in which eight out of nine Evos were highly significant (Fig. 1B, upper). However, the magnitudes of growth improvement were considerably varied, and the evolutionary dynamics of the nine lineages were somehow divergent (Fig. S2).”

(5) The evolved lineages showed a decrease in their maximal optical densities (OD600) compared to the ancestral strain (L85-86). ALE could accompany changes in cell size and morphologies, (doi: 10.1038/s41586-023-06288-x; 10.1128/AEM.01120-17), which may render OD600 relatively inaccurate for cell density comparison. I suggest using CFU/mL metrics for the sake of a fair comparison between Anc and Evo.

The methods evaluating the carrying capacity (i.e., cell density, population size, etc.) do not change the results. Even using CFU is unfair for the living cells that can not form colonies and unfair if the cell size changes. Optical density (OD600) provides us with the temporal changes of cell growth in a 15-minute interval, which results in an exact evaluation of the growth rate in the exponential phase. CFU is poor at recording the temporal changes of population changes, which tend to result in an inappropriate growth rate. Taken together, we believe that our method was reasonable and reliable. We hope you can accept the different way of study.

(6) Please provide evidence in support of the statement in L115-119. i.e. statistical analysis supporting that the observed ratio of essential genes in the mutant pool is not random.

The statistic test was performed, and the following sentence was added.

(L139-141) “The ratio of essential genes in the mutated genes was significantly higher than in the total genes (286 out of 3290 genes, Chi-square test p=0.008).”

(7) The assumption that "mutation abundance would correlate to fitness improvement" described in L120-122: "The large variety in genome mutations and no correlation of mutation abundance to fitness improvement strongly suggested that no mutations were specifically responsible or crucially essential for recovering the growth rate of the reduced genome" is not easy to digest, in the sense that (i) the effect of multiple beneficial mutations are not necessarily summative, but are riddled with various epistatic interactions (doi: 10.1016/j.mec.2023.e00227); (ii) neutral hitchhikers are of common presence (you could easily find reference on this one); (iii) hypermutators that accumulate greater number of mutations in a given time are not always the eventual winners in competition games (doi: 10.1126/science.1056421). In this sense, the notion that "mutation abundance correlates to fitness improvement" in L120-122 seems flawed (for your perusal, doi: 10.1186/gb-2009-10-10-r118).

Sorry for the improper description and confusing writing, and thank you for the fruitful knowledge on molecular evolution. The sentence was deleted, and the following one was added.

(L145-146) “Nevertheless, it was unclear whether and how these mutations were explicitly responsible for recovering the growth rate of the reduced genome.”

(8) Could it be possible that the large variation in genome mutations in independent lineages results from a highly rugged fitness landscape characterized by multiple fitness optima (doi: 10.1073/pnas.1507916112)? If this is the case, I disagree with the notion in L121-122 "that no mutations were specifically responsible or crucially essential" It does seem to me that, for example, the mutations in evo A2 are specifically responsible and essential for the fitness improvement of evo A2 in the evolutionary condition (M63 medium). Fitness assessment of individual (or combinatorial) mutants reconstituted in the Ancestral background would be a bonus.

Thank you for the intriguing thinking. The sentence was deleted. Please allow us to adapt your comment to the manuscript as follows.

(L143-145) “The large variety of genome mutations fixed in the independent lineages might result from a highly rugged fitness landscape 38.”

(9) L121-122: "...no mutations were specifically responsible or crucially essential for recovering the growth rate of the reduced genome". Strictly speaking, the authors should provide a reference case of wild-type *E. coli* ALE in order to reach definitive conclusions that the observed mutation events are exclusive to the genome-reduced strain. It is strongly recommended that the authors perform comparative analysis with an ALEed non-genome-reduced control for a more definitive characterization of the evolutionary biology in a genome-reduced organism, as it was done for "JCVI-syn3.0B vs non-minimal M. mycoides" (doi: 10.1038/s41586-023-06288-x) and "*E. coli* eMS57 vs MG1655" (doi: 10.1038/s41467-019-08888-6).

The improper description was deleted in response to comments 7 and 8. The mentioned references were cited in the manuscript (refs 21 and 23). Thank you for the experimental advice. We are sorry that the comparison of wild-type and reduced genomes was not in the scope of the present study and will probably be reported soon in our future work.

(10) L146-148: "The homeostatic periodicity was consistent with our previous findings that the chromosomal periodicity of the transcriptome was independent of genomic or environmental variation" A Previous study also suggested that the amplitudes of the periodic transcriptomes were significantly correlated with the growth rates (doi: 10.1093/dnares/dsaa018). Growth rates of 8/9 Evos were higher compared to Anc, while that of Evo F2 remained similar. Please comment on the changes in amplitudes of the periodic transcriptomes between Anc and each Evo.

Thank you for the suggestion. The correlation between the growth rates and the amplitudes of chromosomal periodicity was statistically insignificant (p>0.05). It might be a result of the limited data points. Compared with the only nine data points in the present study, the previous study analyzed hundreds of transcriptomes associated with the corresponding growth rates, which are suitable for statistical evaluation. In addition, the changes in growth rates were more significant in the previous study than in the present study, which might influence the significance. It's why we did not discuss the periodic amplitude.

(11) Please elaborate on L159-161: "It strongly suggested the essentiality mutation for homeostatic transcriptome architecture happened in the reduced genome.".

Sorry for the improper description. The sentence was rewritten as follows.

(L191-193) “The essentiality of the mutations might have participated in maintaining the homeostatic transcriptome architecture of the reduced genome.”

(12) Is FPKM a valid metric for between-sample comparison? The growing consensus in the community adopts Transcripts Per Kilobase Million (TPM) for comparing gene expression levels between different samples (Figure 3B; L372-379).

Sorry for the unclear description. The FPKM indicated here was globally normalized, statistically equivalent to TPM. The following sentence was added to the Materials and Methods.

(L421-422) “The resulting normalized FPKM values were statistically equivalent to TPM.”

(13) Please provide % mapped frequency of mutations in Table S3.

They were all 100%. The partially fixed mutations were excluded in the present study. The following sentence was added to the caption of Table S3.

(Supplementary file, p 9) “Note that the entire population held the mutations, i.e., 100% frequency in DNA sequencing.”

(14) To my knowledge, M63 medium contains glucose and glycerol as carbon sources. The manuscript would benefit from discussing the elements that impose selection pressure in the M63 culture condition.

Sorry for the missing information on M63, which contains 22 mM glucose as the only carbon source. The medium composition was added in the Materials and Methods, as follows.

(L334-337) “In brief, the medium contains 62 mM dipotassium hydrogen phosphate, 39 mM potassium dihydrogen phosphate, 15 mM ammonium sulfate, 15 μM thiamine hydrochloride, 1.8 μM Iron (II) sulfate, 0.2 mM magnesium sulfate, and 22 mM glucose.”

(15) The RNA-Seq datasets for Evo strains seemed equally heterogenous, just as their mutation profiles. However, the missing element in their analysis is the directionality of gene expression changes. I wonder what sort of biological significance can be derived from grouping expression changes based solely on DEGs, without considering the magnitude and the direction (up- and down-regulation) of changes? RNA-seq analysis in its current form seems superficial to derive biologically meaningful interpretations.

We agree that most studies often discuss the direction of transcriptional changes. The present study aimed to capture a global view of the magnitude of transcriptome reorganization. Thus, the analyses focused on the overall features, such as the abundance of DEGs, instead of the details of the changes, e.g., the up- and down-regulation of DEGs. The biological meaning of the DEGs' overview was how significantly the genome-wide gene expression fluctuated, which might be short of an in-depth view of individual gene expression. The following sentence was added to indicate the limitation of the present analysis.

(L199-202) “Instead of an in-depth survey on the directional changes of the DEGs, the abundance and functional enrichment of DEGs were investigated to achieve an overview of how significant the genome-wide fluctuation in gene expression, which ignored the details of individual genes.”

Minor remarks(1) L41: brackets italicized "(*E. coli*)".

It was fixed as follows.

(L40) “… *Escherichia coli* (*E. coli*) cells …”

(2) Figure S1. It is suggested that the x-axis of ALE monitor be set to 'generations' or 'cumulative generations', rather than 'days'.

Thank you for the suggestion. Fig. S1 describes the experimental procedure, so the" day" was used. Fig. S2 presents the evolutionary process, so the "generation" was used, as you recommended here.

(3) I found it difficult to digest through L61-64. Although it is not within the job scope of reviewers to comment on the language style, I must point out that the manuscript would benefit from professional language editing services.

Sorry for the unclear writing. The sentences were revised as follows.

(L60-64) “Previous studies have identified conserved features in transcriptome reorganization, despite significant disruption to gene expression patterns resulting from either genome reduction or experimental evolution 27-29. The findings indicated that experimental evolution might reinstate growth rates that have been disrupted by genome reduction to maintain homeostasis in growing cells.”

(4) Duplicate references (No. 21, 42).

Sorry for the mistake. It was fixed (leaving ref. 21).

(5) Inconsistency in L105-106: "from two to 13".

"From two to 13" was adopted from the language editing. It was changed as follows.

(L119) “… from 2 to 13, …”

**Response to Reviewer #3:**

Thank you for reviewing our manuscript and for the helpful comments, which improved the strength of the manuscript. The recommended statistical analyses essentially supported the statement in the manuscript were performed, and those supposed to be the new results in the scope of further studies remained unconducted. The changes made in the revision were highlighted. We sincerely hope the revised manuscript and the following point-to-point response meet your concerns. You will find all your suggested statistic tests in our future work that report an extensive study on the experimental evolution of an assortment of reduced genomes.

(1) Line 106 - "As 36 out of 45 SNPs were nonsynonymous, the mutated genes might benefit the fitness increase." This argument can be strengthened. For example, the null expectation of nonsynonymous SNPs should be discussed. Is the number of observed nonsynonymous SNPs significantly higher than the expected one?(2) Line 107 - "In addition, the abundance of mutations was unlikely to be related to the magnitude of fitness increase." Instead of just listing examples, a regression analysis can be added.

Yes, it's significant. Random mutations lead to ~33% of nonsynonymous SNP in a rough estimation. Additionally, the regression is unreliable because there's no statistical significance between the number of mutations and the magnitude of fitness increase. Accordingly, the corresponding sentences were revised with additional statistical tests.

(L123-129) “As 36 out of 45 SNPs were nonsynonymous, which was highly significant compared to random mutations (p < 0.01), the mutated genes might benefit fitness increase. In addition, the abundance of mutations was unlikely to be related to the magnitude of fitness increase. There was no significant correlation between the number of mutations and the growth rate in a statistical view (p > 0.1). Even from an individual close-up viewpoint, the abundance of mutations poorly explained the fitness increase.”

(3) Line 114 - "They seemed highly related to essentiality, as 11 out of 49 mutated genes were essential (Table S3)." Here, the information mentioned in line 153 ("the ratio of essential to all genes (302 out of 3,290) in the reduced genome.") can be used. Then a statistical test for a contingency table can be used.(4) Line 117 - "the high frequency of the mutations fixed in the essential genes suggested the mutation in essentiality for fitness increase was the evolutionary strategy for reduced genome." What is the expected number of fixed mutations in essential genes vs non-essential genes? Is the observed number statistically significantly higher?

Sorry for the improper and insufficient information on the essential genes. Yes, it's significant. The statistical test was additionally performed. The corresponding part was revised as follows.

(L134-146) “They seemed highly related to essentiality7 (https://shigen.nig.ac.jp/ecoli/pec/genes.jsp), as 11 out of 49 mutated genes were essential (Table S3). Although the essentiality of genes might differ between the wild-type and reduced genomes, the experimentally determined 302 essential genes in the wild-type *E. coli* strain were used for the analysis, of which 286 were annotated in the reduced genome. The ratio of essential genes in the mutated genes was significantly higher than in the total genes (286 out of 3290 genes, Chi-square test p=0.008). As the essential genes were determined according to the growth35 and were known to be more conserved than nonessential ones 36,37, the high frequency of the mutations fixed in the essential genes was highly intriguing and reasonable. The large variety of genome mutations fixed in the independent lineages might result from a highly rugged fitness landscape 38. Nevertheless, it was unclear whether and how these mutations were explicitly responsible for recovering the growth rate of the reduced genome.”

(5) The authors mentioned no overlapping in the single mutation level. Is that statistically significant? The authors can bring up what the no-overlap probability is given that there are in total x number of fixed mutations observed (either theory or simulation is good).

Sorry, we feel confused about this comment. It's unclear to us why it needs to be statistically simulated. Firstly, the mutations were experimentally observed. The result that no overlapped mutated genes were detected was an Experimental Fact but not a Computational Prediction. We feel sorry that you may over-interpret our finding as an evolutionary rule, which always requires testing its reliability statistically. We didn't conclude that the evolution had no overlapped mutations. Secondly, considering 65 times random mutations happened to a ~3.9 Mb sequence, the statistical test was meaningful only if the experimental results found the overlapped mutations. It is interesting how often the random mutations cause the overlapped mutations in parallel evolutionary lineages while increasing the evolutionary lineages, which seems to be out of the scope of the present study. We are happy to include the analysis in our ongoing study on the experimental evolution of reduced genomes.

(6) The authors mentioned no overlapping in the single mutation level. How about at the genetic level? Some fixed mutations occur in the same coding gene. Is there any gene with a significantly enriched number of mutations?

No mutations were fixed in the same gene of biological function, as shown in Table S3. If we say the coding region, the only exception is the IS sequences, well known as the transposable sequences without genetic function. The following description was added.

(L119-122) “The number of mutations largely varied among the nine Evos, from 2 to 13, and no common mutation was detected in all nine Evos (Table S3). A 1,199-bp deletion of insH was frequently found in the Evos (Table S3, highlighted), which well agreed with its function as a transposable sequence.”

(7) Line 151-156- It seems like the authors argue that the expression level differences can be just explained by the percentage of essential genes that get fixed mutations. One further step for the argument could be to compare the expression level of essential genes with vs without fixed mutations. Also, the authors can compare the expression level of non-essential genes with vs without fixed mutations. And the authors can report whether the differences in expression level became insignificant after the control of the essentiality.

It's our pleasure that the essentiality intrigued you. Thank you for the analytical suggestion, which is exciting and valuable for our studies. As only 11 essential genes were detected here and "Mutation in essentiality" was an indication but not the conclusion of the present study, we would like to apply the recommended analysis to the datasets of our ongoing study to demonstrate this statement. Thank you again for your fruitful analytical advice.

(8) Line 169- "The number of DEGs partially overlapped among the Evos declined significantly along with the increased lineages of Evos (Figure 4B). " There is a lack of statistical significance here while the word "significantly" is used. One statistical test that can be done is to use re-sampling/simulation to generate a null expectation of the overlapping numbers given the DEGs for each Evo line and the total number of genes in the genome. The observed number can then be compared to the distribution of the simulated numbers.

Sorry for the inappropriate usage of the term. Whether it's statistically significant didn't matter here. The word "significant" was deleted as follows.

(L205--206) “The number of DEGs partially overlapped among the Evos declined along with the increased lineages of Evos (Fig. 4B).”

(9) Line 177-179- "In comparison,1,226 DEGs were induced by genome reduction. The common DEGs 177 of genome reduction and evolution varied from 168 to 540, fewer than half of the DEGs 178 responsible for genome reduction in all Evos" Is the overlapping number significantly lower than the expectation? The hypergeometric test can be used for testing the overlap between two gene sets.

There's no expectation for how many DEGs were reasonable. Not all numbers experimentally obtained are required to be statistically meaningful, which is commonly essential in computational and data science.

(10) The authors should give more information about the ancestral line used at the beginning of experimental evolution. I guess it is one of the KHK collection lines, but I can not find more details. There are many genome-reduced lines. Why is this certain one picked?

Sorry for the insufficient information on the reduced genome used for the experimental evolution. The following descriptions were added in the Results and the Materials and Methods, respectively.

(L75-79) “The *E. coli* strain carrying a reduced genome, derived from the wild-type genome W3110, showed a significant decline in its growth rate in the minimal medium compared to the wild-type strain 13. To improve the genome reduction-mediated decreased growth rate, the serial transfer of the genome-reduced strain was performed with multiple dilution rates to keep the bacterial growth within the exponential phase (Fig. S1), as described 17,20.”

(L331-334) “The reduced genome has been constructed by multiple deletions of large genomic fragments 58, which led to an approximately 21% smaller size than its parent wild-type genome W3110.”

(11) How was the saturated density in Figure 1 actually determined? In particular, the fitness assay of growth curves is 48h. But it seems like the experimental evolution is done for ~24 h cycles. If the Evos never experienced a situation like a stationary phase between 24-48h, and if the author reported the saturated density 48 h in Figure 1, the explanation of the lower saturated density can be just relaxation from selection and may have nothing to do with the increase of growth rate.

Sorry for the unclear description. Yes, you are right. The evolution was performed within the exponential growth phase (keeping cell division constant), which means the Evos never experienced the stationary phase (saturation). The final evolved populations were subjected to the growth assay to obtain the entire growth curves for calculating the growth rate and the saturated density. Whether the decreased saturated density and the increased growth rate were in a trade-off relationship remained unclear. The corresponding paragraph was revised as follows.

(L100-115) “Intriguingly, a positive correlation was observed between the growth fitness and the carrying capacity of the Evos (Fig. 1D). It was somehow consistent with the positive correlations between the colony growth rate and the colony size of a genome-reduced strain 11 and between the growth rates and the saturated population size of an assortment of genome reduced strains 13. Nevertheless, the negative correlation between growth rate and carrying capacity, known as the r/K selection30,31 was often observed as the trade-off relationship between r and K in the evolution and ecology studies 32 33,34. As the r/K trade-off was proposed to balance the cellular metabolism that resulted from the cost of enzymes involved 34, the deleted genes might play a role in maintaining the metabolism balance for the r/K correlation. On the other hand, the experimental evolution (i.e., serial transfer) was strictly performed within the exponential growth phase; thus, the evolutionary selection was supposed to be driven by the growth rate without selective pressure to maintain the carrying capacity. The declined carrying capacity might have been its neutral "drift" but not a trade-off to the growth rate. Independent and parallel experimental evolution of the reduced genomes selecting either r or K is required to clarify the actual mechanisms.”

(12) What annotation of essentiality was used in this paper? In particular, the essentiality can be different in the reduced genome background compared to the WT background.

Sorry for the unclear definition of the essential genes. They are strictly limited to the 302 essential genes experimentally determined in the wild-type E coli strain. Detailed information can be found at the following website: https://shigen.nig.ac.jp/ecoli/pec/genes.jsp. We agree that the essentiality could differ between the WT and reduced genomes. Identifying the essential genes in the reduced genome will be an exhaustedly vast work. The information on the essential genes defined in the present study was added as follows.

(L134-139) “They seemed highly related to essentiality7 (https://shigen.nig.ac.jp/ecoli/pec/genes.jsp), as 11 out of 49 mutated genes were essential (Table S3). Although the essentiality of genes might differ between the wild-type and reduced genomes, the experimentally determined 302 essential genes in the wild-type *E. coli* strain were used for the analysis, of which 286 were annotated in the reduced genome.”

(13) The fixed mutations in essential genes are probably not rarely observed in experimental evolution. For example, fixed mutations related to RNA polymerase can be frequently seen when evolving to stressful environments. I think the author can discuss this more and elaborate more on whether they think these mutations in essential genes are important in adaptation or not.

Thank you for your careful reading and the suggestion. As you mentioned, we noticed that the mutations in RNA polymerases (rpoA, rpoB, and rpoD) were identified in three Evos. As they were not shared across all Evos, we didn't discuss the contribution of these mutations to evolution. Instead of the individual functions of the mutated essential gene functions, we focused on the enriched gene functions related to the transcriptome reorganization because they were the common feature observed across all Evos and linked to the whole metabolic or regulatory pathways, which are supposed to be more biologically reasonable and interpretable. The following sentence was added to clarify our thinking.

(L268-273) “In particular, mutations in the essential genes, such as RNA polymerases (rpoA, rpoB, rpoD) identified in three Evos (Table S3), were supposed to participate in the global regulation for improved growth. Nevertheless, the considerable variation in the fixed mutations without overlaps among the nine Evos (Table 1) implied no common mutagenetic strategy for the evolutionary improvement of growth fitness.”

(14) In experimental evolution to new environments, several previous literature also show that long-term experimental evolution in transcriptome is not consistent or even reverts the short-term response; short-term responses were just rather considered as an emergency plan. They seem to echo what the authors found in this manuscript. I think the author can refer to some of those studies more and make a more throughput discussion on short-term vs long-term responses in evolution.

Thank you for the advice. It's unclear to us what the short-term and long-term responses referred to mentioned in this comment. The "Response" is usually used as the phenotypic or transcriptional changes within a few hours after environmental fluctuation, generally non-genetic (no mutation). In comparison, long-term or short-term experimental "Evolution" is associated with genetic changes (mutations). Concerning the Evolution (not the Response), the long-term experimental evolution (>10,000 generations) was performed only with the wild-type genome, and the short-term experimental evolution (500~2,000 generations) was more often conducted with both wild-type and reduced genomes, to our knowledge. Previous landmark studies have intensively discussed comparing the wild-type and reduced genomes. Our study was restricted to the reduced genome, which was constructed differently from those reduced genomes used in the reported studies. The experimental evolution of the reduced genomes has been performed in the presence of additional additives, e.g., antibiotics, alternative carbon sources, etc. That is, neither the genomic backgrounds nor the evolutionary conditions were comparable. Comparison of nothing common seems to be unproductive. We sincerely hope the recommended topics can be applied in our future work.

Some minor suggestionsFigures S3 & Table S2 need an explanation of the abbreviations of gene categories.

Sorry for the missing information. Figure S3 and Table S3 were revised to include the names of gene categories. The figure was pasted followingly for a quick reference.

**Author response image 3. sa4fig3:** 

I hope the authors can re-consider the title; "Diversity for commonality" does not make much sense to me. For example, it can be simply just "Diversity and commonality."

Thank you for the suggestion. The title was simplified as follows.

(L1) “Experimental evolution for the recovery of growth loss due to genome reduction.”

It is not easy for me to locate and distinguish the RNA-seq vs DNA-seq files in DRA013662 at DDBJ. Could you make some notes on what RNA-seq actually are, vs what DNA-seq files actually are?

Sorry for the mistakes in the DRA number of DNA-seq. DNA-seq and RNA-seq were deposited separately with the accession IDs of DRA013661 and DRA013662, respectively. The following correction was made in the revision.

(L382-383) “The raw datasets of DNA-seq were deposited in the DDBJ Sequence Read Archive under the accession number DRA013661.”